# Firmware Updates over the Air via LoRa: Unicast and Broadcast Combination for Boosting Update Speed

**DOI:** 10.3390/s24072104

**Published:** 2024-03-25

**Authors:** Victor Malumbres, Jose Saldana, Gonzalo Berné, Julio Modrego

**Affiliations:** 1CIRCE Technology Center, Avenida Ranillas, 50018 Zaragoza, Spain; vmalumbres@fcirce.es (V.M.); gberne@fcirce.es (G.B.); 2Airfal International, C. Río Ésera, 5, Villanueva de Gállego, 50830 Zaragoza, Spain

**Keywords:** IoT, IIoT, LoRa, cybersecurity, over the air update, FUOTA, ATEX, firmware update

## Abstract

The capacity to update firmware is a vital component in the lifecycle of Internet of Things (IoT) devices, even those with restricted hardware resources. This paper explores the best way to wirelessly (Over The Air, OTA) update low-end IoT nodes with difficult access, combining the use of unicast and broadcast communications. The devices under consideration correspond to a recent industrial IoT project that focuses on the installation of intelligent lighting systems within ATEX (potentially explosive atmospheres) zones, connected via LoRa to a gateway. As energy consumption is not limited in this use case, the main figure of merit is the total time required for updating a project. Therefore, the objective is to deliver all the fragments of the firmware to each and all the nodes in a safe way, in the least amount of time. Three different methods, combining unicast and broadcast transmissions in different ways, are explored analytically, with the aim of obtaining the expected update time. The methods are also tested via extensive simulations, modifying different parameters such as the size of the scenario, the number of bytes of each firmware chunk, the number of nodes, and the number of initial broadcast rounds. The simulations show that the update time of a project can be significant, considering the limitations posed by regulations, in terms of the percentage of airtime consumption. However, significant time reductions can be achieved by using the proper method: in some cases, when the number of nodes is high, the update time can be reduced by two orders of magnitude if the correct method is chosen. Moreover, one of the proposed methods is implemented using actual hardware. This real implementation is used to perform firmware update experiments in a lab environment. Overall, the article illustrates the advantage of broadcast approaches in this kind of technology, in which the transmission rate is constant despite the distance between the gateway and the node. However, the advantage of these broadcast methods with respect to the unicast one could be mitigated if the nodes do not run exactly the same firmware version, since the control of the broadcast update would be more difficult and the total update time would increase.

## 1. Introduction

The ability to perform firmware updates is an essential part of the lifecycle of Internet of Things (IoT) devices, even those with constrained hardware resources. This is primarily because security algorithms require regular updates, either as a response to the discovery of a vulnerability or as a proactive measure. Additionally, firmware updates facilitate the introduction of new functionalities on devices.

In certain situations, devices requiring updates can be manually flashed using a connector. Alternatively, they may be connected via a cable that allows for the update. However, in many instances, these devices are inaccessible. For example, they may be in the field or situated in ATEX (potentially explosive atmospheres) zones, where access requires specially trained personnel and specific equipment. For instance, a luminaire designed for an ATEX zone might be installed on the ceiling of an oil and gas facility’s warehouse, making it challenging to reach. In addition, the installation of additional wiring may not be a feasible solution in these scenarios. Many industrial IoT (IIoT) devices are dispersed across the field, and, particularly in ATEX zones, regulations restrict the installation of extra wires. At the same time, security is a must in these areas, usually related to critical infrastructures which require relatively frequent security updates.

Many IoT protocols have been designed with the main objective of having a long range, at the cost of a limited bandwidth. From the lowest to the upper layers, they are conceived with a specific purpose in mind: to enable the efficient transmission of sensor data over extensive distances, while necessitating only a minimal byte count. In many instances, these protocols function within license-free bands of the electromagnetic spectrum. Regulatory bodies impose certain restrictions on airtime usage to prevent potential congestion situations. Consequently, the volume of information that can be transmitted by each device per unit of time is significantly limited.

Numerous wireless protocols possess the capability to transmit both broadcast (or multicast) and unicast messages to and from devices. Although broadcast messages may, in principle, provide a higher level of scalability, they have a limitation: they cannot be acknowledged, so the sender is unable to ascertain which devices have received the messages and which have not. Furthermore, the sending of broadcast messages cannot be optimized (rate and power) for each of the receivers.

In this context, the problem to be addressed in this paper is to explore the ways to wirelessly (Over The Air, OTA) update low-end IoT nodes with difficult access, combining the use of broadcast and unicast communications. Specifically, the devices used in the present study would correspond to the *Low-end*, *Class 2* category of the classification presented in [1] (roughly 50 kB of RAM and 250 kB of flash). The challenge is to streamline the roll out of a system update, eliminating the need for physical interaction with the device, while minimizing the time required. The update requires the successful transmission of a set of fragments (chunks) of the firmware, and the subsequent flashing of the received update by each and all the nodes.

If the quality of communication is high, an initial transmission of all firmware chunks via broadcast/multicast frames could lead to a significant proportion of nodes possessing a large portion of these fragments. Following this broadcast phase, a unicast communication could be used to ascertain the status of each node and deliver any remaining segments. However, in situations where communication quality is poor, the initial broadcast phase may not be as effective. This raises several research questions related to scalability: Is it always beneficial to start with a broadcast phase? Would it be more effective to conduct multiple broadcast phases prior to the final unicast stage? Other questions of interest are: Should the maximum frame size always be used? How does the number of nodes modify the problem?

As we will see, some architectures and methods for IoT updates [2] have been proposed, which are divided into different stages, such as, e.g., the validation of a manifest, the firmware exchange, the verification of a checksum, and the final secure flashing. It should be noted that the present paper is mainly focused on the firmware exchange stage: the phase in which the firmware chunks are sent through the air from the gateway to each of the devices to be updated.

The devices under consideration correspond to a recent project that focuses on the installation of intelligent lighting systems within ATEX zones. The nodes are low-end IoT devices equipped with a microcontroller unit (MCU) of limited capacity, which is capable of collecting data regarding light (consumption metrics, temperature readings, and any detected malfunctions) using the DALI (Digital Addressable Lighting Interface) standard [3]. Furthermore, it is also capable of gathering data from connected sensors. More specific information about the devices is provided in Section 6.

As far as communications are concerned, LoRa is chosen for its extensive adoption as a wireless protocol that possesses all the necessary attributes: in the considered industrial environments, it has good penetration, because it works in the license-free sub-gigahertz band. In addition, ATEX elements require an enclosure which absorbs some amount of energy. As we will see, for different reasons, we opt for the use of MiWi [4,5] above the physical LoRa layer, instead of resorting to the more commonly used LoRaWAN [6,7]. Although the experiments will be run on a scenario of this kind (LoRa in the EU868 band, in an ATEX scenario), many of the presented findings may also be valid for other ones.

The main KPIs (Key Performance Indicators) usually considered for OTA update methods are the update time, the energy consumption, and the update efficiency (the percentage of successfully updated nodes) [8]. In the present case, the energy cost incurred by the firmware update is not relevant, since all the nodes are part of ATEX luminaires connected to the power supply network, so batteries are not required. Another requisite is that the update efficiency must be 100%, i.e., the procedure will not finish until all the nodes have been updated. This leaves the update time as our main KPI. We define it as the time required for updating a whole project, including a number of IoT devices, assuming that the update is completed safely (i.e., the correct firmware is flashed on the device). A reduction in the total update time is also relevant from a cybersecurity point of view: if a new vulnerability is disclosed, the devices must be updated as soon as possible. The total time is closely related to the consumed airtime, considering the limitations imposed by regulations. As an example, the LoRa Alliance limits airtime usage to 1% of the total time, and in certain cases, to 0.1% or 10% of it [9]. As we will see, one approach can be to minimize the total update time, while keeping the airtime usage just below the limit. Therefore, the results will be presented in this paper in terms of the total update time.

All in all, the contribution of the paper can be summarized as:

The proposal, definition, and analysis of three methods for OTA firmware updates: an *only unicast* one; another one that combines a number of initial broadcast stages with a final unicast one, in which the pending chunks are sent individually to each node (it will be called *broadcast + unicast*); and an improved one, in which the firmware chunks are always sent in broadcast frames (*only broadcast*).An extensive evaluation of the different methods via simulation, considering a LoRa scenario in the 868 MHz band. A test battery varying different parameters is carried out.The implementation and evaluation, with real hardware, of the *only unicast* method in an IoT intelligent lighting system.

The rest of the paper is organized as follows: Section 2 details the related work; Section 3 presents the three proposed methods; Section 4 includes an analysis of the required update times of each of them; In Section 5, a battery of simulations is presented, tuning different parameters; Section 6 details the implementation of the unicast method in real hardware; and the paper ends with the conclusions and future research.

## 2. Related Work

In this section, we summarize the related research papers regarding OTA updates, security requirements for IoT devices, LoRa MAC protocols, and, finally, simulation tools that can be used to test the considered protocols and scenarios.

### 2.1. Over the Air Update of IoT Devices

Back in 2006, before the rise of the IoT, the authors of [10], when talking about the updating of wireless sensor networks, detailed the main challenges to be met by firmware update algorithms: they should adapt to the limited processing capacity and memory of the devices and they should be energy efficient, while ensuring the correct delivery of the firmware. Finally, scalability was also considered as a relevant question, since it was (rightly) expected that node density could increase dramatically in the near future.

More recently, an examination was carried out on the support for OTA updates across 26 open-source IoT operating systems and embedded software projects [11]. The findings revealed that comprehensive information regarding the implementation of OTA updates was not provided by all projects, so the authors advocated for the establishment of standard mechanisms. This would eliminate the necessity to construct OTA update systems from scratch each time a new technology is under consideration for testing or implementation.

A recent standardization effort carried out by the Software Updates for Internet of Things (SUIT) Working Group of the Internet Engineering Task Force (IETF) resulted in the proposal of an architecture for firmware updates applicable to IoT devices, as detailed in [2]. The architecture defines a set of entities: the cloud part, a firmware server, and the end devices. It also defines a set of stages, including the validation of a manifest, the firmware exchange, its validation, and secure flashing, etc.

The authors of [12] pioneered the implementation and testing of a firmware update solution aligned with the architecture proposed by the IETF. Their tests were conducted in two distinct environments, one of them using Wi-Fi in an LAN, and the other in a real-world scenario with LoRaWAN gateways. Carlson’s work [13] was constructed entirely on the foundation of the same IETF architecture, prior to its formalization as in [2], demonstrating its adaptability and practicality. The implementation used Contiki-NG [14] as the embedded operating system and the tests were carried out on a single board device.

The study in [15] presented LoRaP2P, describing an architecture and a method for firmware updates in agricultural scenarios. A TDMA-based (Time-Division Multiple Access) MAC protocol was defined, including a cycle of 32 time slots of 0.5 s each. In the *firmware updating* stage, the gateway sent the chunks in broadcast frames, including 30 bytes of the firmware each, according to a customized packet structure. A “go-back N” mechanism was implemented. To avoid problems if a node was restarted, the received chunks were stored in a non-volatile memory. The system was implemented and tested with real hardware.

The authors of [16] presented a system for remotely flashing ATMEL AVR microcontrollers via Wi-Fi and LoRa technologies. The paper was mainly focused on the hardware part, and not on the exchange of firmware chunks.

A simulation tool called FUOTASim was developed in [8], with the aim of studying the effects of LoRaWAN parameters on the firmware update process, considering the impact on time, energy consumption, and efficiency. It allows the use of different data rates, firmware and chunk sizes, and redundant codes. The study presented in [17] proposed a firmware update protocol over LoRaWAN based on adaptative data rate techniques, defined as a couple of spreading factor and bandwidth values. It was studied by the means of simulations, using as KPIs the energy consumption of the end devices powered by batteries and the security. The possibility of the firmware being stolen during the FUOTA process was also studied.

The present paper is focused on the firmware delivery stage, and not so much on the architecture (although an architecture is defined and employed, presented in Section 6). In contrast to [15], in our scenario, the use of a TDMA protocol is not feasible due to two specific reasons: first, it would require a synchronization mechanism; and second, in our system, there are some asynchronous events that cannot wait for much time. To the best of our knowledge, there is no standard procedure for firmware exchange in an OTA update over LoRa. In our study, the focus is on the comparison between three different modes that can be employed in firmware delivery. This happens during a concrete stage of the update, but it is usually the most time-consuming one, as it is carried out using a very slow wireless technology.

### 2.2. Security Update Requirements for IoT devices

In recent years, cyber attacks have become a major problem for industries and IT services. They have the potential to inflict significant harm on businesses, products, and data. In [18], a summary of the major attacks that have occurred in the last 20 years against critical infrastructures was presented, also including an analysis of the types of attacks, consequences, vulnerabilities, victims, and attackers. The paper also provided an estimation of the number of major cyber attacks that will occur on critical infrastructure in the future.

These attacks are continually evolving, growing increasingly sophisticated. Now, with the advent of the IoT, a new paradigm is emerging. IoT devices, ranging from smart home gadgets to industrial equipment, are small and physically isolated, but connected to a larger network, creating a tempting doorway for malicious actors. Many of these devices are responsible for handling sensitive personal data and performing critical tasks, making a loss of information or control a real concern [19]. As they are part of a larger system, individual deep checks of the devices are not performed very frequently, and this fact can be used by attackers to hide hacked devices from which they can read information, monitor workflows, influence system behavior, or communicate with external agents. With this in mind, it is important to establish cybersecurity strategies that keep devices updated in advance of any threat.

Firmware serves as the foundational software embedded within IoT devices. Ensuring its security is essential to prevent unauthorized access, data breaches, and the potential manipulation of devices for malicious purposes. A recent review [20] emphasized the importance of addressing vulnerabilities in firmware to enhance the security posture of IoT ecosystems, thereby reinforcing trust and reliability for stakeholders.

The authors of [21] conducted a comprehensive analysis of the components within an IoT operating system that receive the most updates post-deployment. They offered an exhaustive quantification of the energy consumption tied to each stage of the process. In [22], the authors present the barriers and cybersecurity targets of IoT devices and define an architectural concept for securing Over The Air updates. In both articles, the main reasons for software updates were similar: protocol and standard version updates; efficiency improvement; critical bug fixes and security updates; additional functionalities; integration with third-party IoT systems; or adopting new communication standards and protocols.

Sometimes, devices are allocated in remote and inaccessible locations, especially in industrial IoT (IIoT) applications, highlighting the importance of protecting communications, particularly in critical processes such as FUOTA (Firmware Update Over The Air). As stated by Catuogno and Galdi [23], ensuring the confidentiality, authenticity, and integrity of the data in these communications is essential. Although data encryption safeguards the firmware transmitted during FUOTA, it does not provide assurance against potential data corruption or unauthorized alterations. This is where data integrity and authenticity verification mechanisms become necessary. A checksum may act as a data integrity guardian, and digital signatures provide a layer of authenticity verification.

As stated before, the present work is focused on the stage in which the firmware is sent through the wireless link. This is a critical moment as far as security is concerned, since the attack surface of a wireless protocol is, in general, higher than that of a wired one: devices can be reached through an expanded area that is potentially accessible by a greater number of systems, and more protocols are exposed due to the need for auto negotiation between the involved devices.

### 2.3. LoRa Protocols: MiWi and LoRaWAN

LoRa is a proprietary radio communication technique for the physical layer (Phy), which expresses each symbol as a cyclic shifted chirp over a predefined frequency interval. Several protocols have been defined as the upper layers of LoRa, such as LoRaWAN, developed by the LoRa Alliance [6,7], which is the most widely used one.

LoRaWAN is a MAC layer protocol which comprises end devices, gateways, and servers, using a star topology. End devices send and receive wireless messages to/from gateways, which, in turn, communicate with the server in both directions. The gateway serves as a translator between LoRa and UDP. The server handles the entire network, processing messages and application data.

There are different types of LoRaWAN uplink and downlink messages, such as *Join-request*, *Join-accept*, and others [6,7]. All messages are encrypted with AES-128. The peer’s handshake and activation can be OTAA (Over The Air Activation) or ABP (Activation By Personalization). OTAA is the most secure and recommended method, due to its dynamic address and security key exchange. In contrast, ABP requires hardcoding a device’s address and security keys in its firmware.

The authors of [24] explored the use of wireless mesh networks based on LoRa Phy as an alternative to LoRaWAN. The paper also included an experiment with sensors across a university campus, covered by a single LoRa gateway, cooperating with a set of LoRa devices acting as routers.

An alternative to LoRaWAN was proposed in [25], introducing several enhancements, such as synchronization capabilities in the nodes for self-initialization and self-maintenance. It also ensures reliable message delivery within scheduled time windows, employs a multi-hop routing protocol to extend communication beyond the transmission range, and dynamically adjusts the spreading factor based on the network conditions.

MiWi, developed by Microchip, is another MAC protocol which works on top of LoRa. Its architecture includes: (a) PAN (Personal Area Network) coordinator/s, elements that start and handle the network, and (b) end devices that connect, send, and receive messages to/from coordinators, building a star, a mesh, or a P2P topology [4,5].

MiWi defines some fixed frames (also encrypted by AES-128), such as *Connection request*, *Connection response*, *Connection removal request*, and *Data request*, etc. The peer’s handshake is easier to perform than in LoRaWAN, but it is less secure: it uses two messages, exchanged between devices with a fixed PAN Coordinator Address (usually hardcoded in the firmware).

In the present paper, MiWi was mainly chosen for architectural reasons. The role of a LoRaWAN gateway is limited to relaying messages between the server and the end devices, but our use case required a smarter gateway able to run the logic of the OTA update, thus allowing a better comparison between the different algorithms. There is also a robustness-related reason: the gateway can coordinate the local devices, even if the connection to the server (or the server itself) fails. It is important to note that the gateway is typically linked to the same electrical circuit as the end devices, implying that both components are likely to experience failure and recovery simultaneously. Finally, the communication latency between the end devices and the gateway is lower than the delay in communicating with the server, thus allowing a slightly faster control.

### 2.4. Simulations and Models for LoRa Environments

Many simulation tools exist that allow repeatable and controlled tests in big scenarios with significant numbers of devices. ns-3 [26] is among the most popular ones, and it is widely used for research purposes in wired and wireless scenarios. It is continuously improved and extended in different ways, and it includes LoRa features. In this subsection, we will summarize some works proposing and testing different LoRa functionalities in ns-3.

The authors of [27] extended the FUOTAsim simulator [8] to study a LoRaWAN scenario. Firmware updates were carried out using multiple gateways instead of one. They studied the benefits in terms of device energy consumption, update efficiency, update time, and number of corrupted and lost fragments.

A LoRaWAN module was developed and tested for ns-3 in [28,29]. It can be used to perform simulations of scenarios including Class A devices (one of the three classes of LoRaWAN devices [30]), gateways, and network servers. It is modular, so it can also be used for LoRa Phy simulations, without the need to implement the whole LoRaWAN stack. This is interesting in our case, since we are using MiWi instead of LoRaWAN, as explained previously.

Some ns-3 signal propagation models have been proposed and tested for urban and city scenarios [31,32]. The results of these papers show that Cost-231 and Okumura-Hata are the most fitted models for city scenarios. In addition, other propagation models for low-population scenarios have been included in ns-3, which were compared and summarized in [33]. This paper showed that Nagakami is one of the most realistic ones. Therefore, this is the model that is employed in the current study, given its suitability for industrial settings.

## 3. Proposed Update Methods

A firmware update is a critical procedure which must be performed carefully. For that reason, after sending the binary file, a checksum must also be sent. Only if the checksum is correct will the new firmware be flashed. Three different methods are defined, which only differ in the firmware exchange stage.

The *only unicast* method is the most straightforward one: it always sends unicast frames, which need an ACK. It employs a “stop and wait” policy, so it only moves to the next chunk when the gateway is sure that the previous one has been received, and it only moves to the next node when the gateway is sure that all the chunks have been correctly received by the previous one. It simplifies the control and the processing burden of the node, with the counterpart of bad scalability: the update time will grow linearly with the number of nodes.

The *broadcast + unicast* method just adds an initial broadcast round (or a number of them) and then runs the *only unicast* method, but only with the missing chunks. It is expected to provide a significant scalability improvement, especially in scenarios with a low frame loss probability. However, if the loss probability is high, the initial broadcast round(s) may not be as effective and the update times will be similar to those of the *only unicast* method. As a counterpart, more intricate control will be necessary: as there are no ACKs in broadcast transmissions, the node will have to transmit to the gateway the frames it is missing. As we will see, a bitmap will be employed for that aim.

Finally, the *only broadcast* method is a variation of the previous one, with the objective of increasing the scalability even more: by sending the missing chunks in broadcast frames, it can be expected that other nodes will also benefit from these transmissions to store some of the pending chunks.

In the next subsections, each method will be explained in detail.

### 3.1. Only Unicast Method

The update process is governed by some variables stored by the node: the first one *(bin_frags)* represents the number of received fragments of the binary file. Another variable *(check_frags)* stores the number of received fragments of the checksum. It must be 0 until all the fragments of the binary file have been received. Finally, another variable stores the current firmware version.

These are the stages of the update (see Figure 1):

(a) *Initial stage*. The update starts when the gateway sends an *OTA RESET REQUEST*. The node then sets both counters to 0 and answers with an *OTA RESET CONFIRMED,* which includes its current firmware version. If the gateway does not receive an answer, a timeout triggers a new *OTA RESET REQUEST*. Once the gateway receives the confirmation, it checks that the firmware version running at the node is different from the one that is going to be transmitted and flashed.

(b) *Sending of the binary*. This uses *OTA BINARY FRAGMENT* frames, which include an identifier *(FRAGMENT NUMBER)*. The gateway sends the fragments, following a “stop and wait” policy: if a fragment with the incorrect identifier arrives, it is discarded and a *WAITING FOR BINARY FRAGMENT* message is sent by the node, specifying the expected fragment. When sending the last fragment, the gateway uses a special message called *OTA BINARY LAST FRAGMENT*. After that, it sends an *OTA STATUS REQUEST,* and the node answers with an *OTA STATUS RESPONSE*, including the field *bin_frags* (number of binary fragments).

(c) *Sending of the checksum.* If the number of binary fragments reported by the node is correct, the gateway starts a similar procedure for sending the checksum, which may require one or more frames. The last fragment of the checksum is sent inside an *OTA CHECKSUM LAST FRAGMENT* frame. This stage concludes when the node sends an *OTA STATUS RESPONSE* frame with the correct value of *check_frags* (number of checksum fragments).

(d) *Checksum validation and firmware flashing*. The gateway waits a time interval to let the node calculate the checksum. Then, it sends a new *OTA STATUS REQUEST*. The node answers with an *OTA STATUS RESPONSE*. If everything is correct, the gateway sends an *OTA FLASH NEW VERSION* order. The node flashes the firmware it has stored and reboots. If the checksum is not correct, the procedure starts again.

(e) *Final confirmation*. The gateway waits a time lapse to let the node flash and reboot. Finally, once the node has associated again, it sends an *OTA STATUS REQUEST* frame. In the response, the node includes its firmware version, which should be the new one.

### 3.2. Broadcast + Unicast Method

This version of the update protocol (see Figure 2) makes use of broadcast LoRa frames. However, two unicast stages are also required, as we will next explain.

Each node stores a variable with its current firmware version number, and a bitmap in which each bit is set if the corresponding chunk has been received.

These are the stages:

(a) *Initial stage (unicast).* The gateway sends a unicast message *OTA BROADCAST START REQUEST* to each of the nodes. The message includes the number of the new firmware version, the number of chunks, and the size of each chunk (note that the last chunk may be smaller than the rest). When a node receives this message, it sets all the bits of the bitmap to 0 and answers with an *OTA BROADCAST START CONFIRMATION.* This message includes the number of the current firmware version running in the node.

(b) *Sending of the binary (broadcast)*. The gateway starts sending broadcast *OTA BINARY FRAGMENT* frames, which also include the identifier of the chunk. When a node receives one of these messages, it stores the received bytes in the corresponding position, calculated from the initial memory position, and the identifier is multiplied by the size of the chunk. It also sets the corresponding bit of the bitmap to 1. There is no acknowledgement. The gateway may perform one or more broadcast rounds. If a node has already received a chunk, it just discards new messages containing the same chunk. After this stage, while some nodes may have received all the chunks, others may have only received a portion of them.

(c) *Sending of the pending chunks (unicast).* The gateway starts with the first node. It sends a message *SEND ME THE BITMAP.* The node answers with a *BITMAP* frame, which specifies the identifier of the missing chunks. The gateway sends all these chunks in unicast frames, following the *only unicast* method, but this only applies to the pending chunks.

After this, the method is equal to the three last stages of the unicast mode (not detailed in Figure 2), namely *Sending of the checksum, Checksum validation and firmware flashing*, and *Final confirmation.*

### 3.3. Only Broadcast Method

There is an improvement that can be applied to the previous method, which modifies stage (c). It mainly consists of sending the pending chunks in broadcast frames (see an example in Figure 3): the gateway sends a *SEND ME THE BITMAP* frame to the node, which answers with a *BITMAP* frame specifying the missing chunks: in this example, chunks #1 (0*x*01) and #542 (0*x*021D) are missing. Then, the gateway sends each of them once, in broadcast frames. In the example, the former is received correctly, but the latter is not. Then, the gateway sends again the *SEND ME THE BITMAP* frame. The node answers with the updated bitmap, in this case, only including fragment #542 as *missing*. The gateway sends it again and then requests the bitmap. In the example, the node answers with an *ALL CHUNKS RECEIVED* frame.

The advantage of this method is that, as the gateway has sent chunk #1 (once) and #542 (twice) in broadcast frames, any other nodes which are missing them will also be able to store them in the corresponding position of the flash memory.

As before, the gateway follows an order, and it will not ask node *j* about the chunks it is missing until node *j* − 1 has received all the chunks. Therefore, the final nodes are more likely to have received the complete firmware by the time the gateway queries them.

## 4. Analytical Expression of the Total Update Time

In this subsection, we include an analysis of the main considered KPI, i.e., the time required to update a project. The problem is to find an expression for the time required to correctly transfer *M* chunks of the firmware to each of the *N* nodes in the project. The analysis will be focused on the main part of the update, i.e., sending the firmware chunks. It should be noted that the airtime, i.e., the time in which the medium is occupied, is different from the total time required for the update, because the duty cycle imposes some limitations (e.g., only 1% of the airtime can be used). Therefore, after sending each frame, the nodes must calculate a time interval in which nothing can be sent to the air [9].

In the analysis, we will consider a fixed transfer rate for the whole project, i.e., the gateway uses the same modulation and bandwidth for all the nodes. In the case of LoRa, this means that the gateway will use a fixed spreading factor and channel bandwidth.

### 4.1. Only Unicast Method

The gateway starts with the update of the first node: it sends the first chunk, waits for a valid ACK, then sends the second chunk, and so on. The process is repeated until the *N*^th^ node is updated.

To correctly calculate the update time, the duty cycle must also be considered, as it affects every frame, including the ACKs. It works this way: let us assume that the duty cycle is *d* (a typical value for *d* is 1, which means a duty cycle of 1%). This means that when a node sends or receives a frame that occupies a certain Time on Air (*ToA*), no frame will be sent during a time equivalent to:(1)$$\left(\frac{100}{d}-1\right)\cdot ToA.$$

As an example, if *ToA* is 313 ms and *d* = 1, the nodes will wait 99 × 313 ms = 30,987 s after being allowed to send a new message.

Therefore, the total transmission time for sending a chunk to a node consists of the time spent during the failed attempts, plus the time required for the final successful transmission. Let *p_j_* be the probability of node *j* not receiving a chunk correctly. If a chunk requires *i* transmissions, this means that it has failed *i* − 1 times (a failure happens with a probability *p_j_*) and arrived one time (this happens with a probability 1 − *p_j_*). So, the average number of times that a chunk will be sent to node *j* can be obtained as the next sum:(2)$${\sum_{i=0}^{\infty }i\left(1-p_{j}\right)p_{j}}^{i-1}={\left(1-p_{j}\right)\sum_{i=0}^{\infty }ip_{j}}^{i-1}.$$

The second term is a power series like $\sum_{i=0}^{\infty }i\cdot p^{i-1}$, which converges for ∣*p*∣ < 1. In this case, the sum of the series is given by the formula $S=1/{\left(1-p\right)}^{2}$. Therefore:(3)$${\left(1-p_{j}\right)\sum_{i=0}^{\infty }ip_{j}}^{i-1}=\left(1-p_{j}\right)\frac{1}{{\left(1-p_{j}\right)}^{2}}=\frac{1}{1-p_{j}}.$$

This means that the number of unsuccessful transmissions before the final (successful) one will be:(4)$$\frac{1}{1-p_{j}}-1=\frac{p_{j}}{1-p_{j}}.$$

Let *ToA_ch_* be the transmission time of a chunk and *ToA_ack_* the transmission time of an ACK. Each unsuccessful transmission takes *ToA* plus the corresponding wait time. The time spent on the transmission of unsuccessful chunks to node *j* will be given by the product of the number of transmissions (4) and the time spent on each of them (1):(5)$$\frac{p_{j}}{1-p_{j}}\cdot \left[{ToA}_{ch}+\left(\frac{100}{d}-1\right)\cdot {ToA}_{ch}\right].$$

The time spent by the final successful transmission has four components: first, the time for the transmission of the chunk (*ToA_ch_*); second, the wait time; third, the time on air of the ACK (*ToA_ack_*); and fourth, the wait time after the ACK:(6)$${ToA}_{ch}+\left(\frac{100}{d}-1\right)\cdot {ToA}_{ch}+{ToA}_{ack}+\left(\frac{100}{d}-1\right)\cdot {ToA}_{ack}.$$

Therefore, the time spent sending *M* chunks to *N* nodes using the *only unicast* method (denoted as *t_uni_*) will be given by the expression:(7)$$t_{\mathrm{uni}}\;=\;M\cdot \frac{100}{d}\cdot \sum_{j=1}^{N}\left(\frac{1}{{1-p}_{j}}\cdot {ToA}_{ch}+{ToA}_{ACK}\right).$$

### 4.2. Broadcast + Unicast Method

This method starts with a set of *B* broadcast rounds, in which all the chunks are sent to the air. Each node stores received chunks and builds a bitmap of missing ones. After that, a unicast round starts, asking the first node about the bitmap containing its missing chunks, sending each of them, and so on.

The time required to broadcast *M* chunks *B* times, considering the duty cycle limitations, will be:(8)$$t_{\mathrm{br}}=M\cdot {\mathrm{ToA}}_{\mathrm{ch}}\;\cdot \frac{100}{d}\cdot B.t$$

The probability that a chunk is not received by node *j* after *B* broadcast rounds will be *p_j_ B*. In the final unicast round, the average number of times that a chunk will be sent to node *j* will be $\frac{1}{1-p_{j}}$, as obtained in (4). 

The time spent in the final unicast rounds of all the *N* nodes can be obtained in a similar way to (8), adding two new terms corresponding to the bitmap request (*ToA_req_*) and response (*ToA_bit_*):(9)$$t_{\mathrm{uni}}=\;M\cdot \frac{100}{d}\cdot \sum_{j=1}^{N}\left[{p_{j}}^{B}\cdot \left(\frac{1}{{1-p}_{j}}\cdot {ToA}_{ch}+{ToA}_{ACK}\right)+{ToA}_{req}+{ToA}_{bit}\right].$$

To obtain a simplified expression, we can make an approximation that all loss probabilities are approximately equal (i.e., *p_j_* ≈ *p*). Therefore, the sum of *N* equal elements can be expressed as a product (by *N*):(10)$$t_{\mathrm{uni}}\approx \;M\cdot \frac{100}{d}\cdot N\cdot \left[p^{B}\cdot \left(\frac{1}{1-p}\cdot {ToA}_{ch}+{ToA}_{ACK}\right)+{ToA}_{req}+{ToA}_{bit}\right].$$

The total time required for *B* broadcast rounds plus a unicast one for each of the *N* nodes can be obtained as the sum of (8) and (10):(11)$$t_{\mathrm{br}+\mathrm{uni}}\;\approx \;M\cdot \frac{100}{d}\cdot \left[{ToA}_{ch}\cdot B+N[p^{B}\cdot \left(\frac{1}{1-p}\cdot {ToA}_{ch}+{ToA}_{ACK}\right)+{ToA}_{req}+{ToA}_{bit}]\right].$$

If we obtain the derivative of *t_br+uni_*, with respect to *B*, and equal it to zero, an optimal value of *B* that minimizes the total time can be obtained. The expression can be simplified assuming that the length of the chunk is much higher than the length of the ACK, i.e., $\frac{1}{1-p}\cdot {ToA}_{ch}\gg {ToA}_{ACK}$:(12)$$B={log}_{p}\left(\frac{p-1}{Nln\left(p\right)}\right).$$

As an example, Figure 4 is obtained with *M* = 534 chunks (23 bytes of header and 192 of payload each, i.e., a firmware of 1024 kB), a Spreading Factor = 7, and different values of *N* and *B*. In Figure 4a, the frame loss probability is 1%. It can be observed that the optimal number of broadcast rounds (*B*) is always 1. However, if the loss probability is significantly higher (40%), this can be different: in Figure 4b, the optimal value of *B* is not 1. Even for 10 nodes, the best option is to run three rounds at the beginning, and the optimal value of *B* grows with the number of nodes: for example, with *N* = 50 nodes, the optimal value is *B* = 5. In both figures, it can be observed that the time cost stabilizes after some broadcast rounds: if the number of nodes is high, the time of a broadcast round is negligible with respect to the time spent in the unicast ones. As a conclusion, when the loss probability is high, it is advisable to perform a number of broadcast rounds before the final unicast one.

### 4.3. Only Broadcast Method

This method has a first stage in which all the chunks are broadcast *B* times. The time required for this stage is the one obtained in (8).

In the second stage, the gateway sends a request to the first node, specifically inquiring about the chunks that are missing. After the node’s response including the bitmap, these chunks are sent in broadcast frames that can also be received by any other node. After that, the gateway asks again about the missing chunks: if the response of the first node is an *ALL CHUNKS RECEIVED* frame, the gateway switches to the second node; otherwise, it means that the first node is still missing some chunks, so they are broadcast again to it, and so on.

To obtain the total update time of a node during the second stage, we must calculate the number of times a chunk is sent to the air in this phase. For simplicity, we will assume that the time required to exchange the bitmaps is negligible with respect to the time required to exchange the firmware chunks (${ToA}_{ch}\gg {ToA}_{req}+{ToA}_{bit}$).

Let *n_i,j_* be the number of times that chunk *i* is sent to node *j*. To calculate its expected value for the first node (*j* = 1), namely E[*n_i_*_,1_], we can consider that it is the product of the probability of node 1 not having received chunk *i* in the first *B* rounds (*p*_1_*^B^*), and the number of times it will be sent specifically to node 1 in its round, i.e., 1/(1 − *p*_1_):(13)$$E[n_{i,1}]\;=\;\frac{1}{1-p_{1}}\cdot {p_{1}}^{B}.$$

The subsequent values of E[*n_i,j_*] (*j* > 1) will depend on the number of times that chunk *i* has been previously sent to the air (E[*n_i,j_*_−1_]). E[*n_i,j_*] can be obtained as the sum of E[*n*_*i,j*−1_] plus the number of times the chunk is sent in this round, i.e., the product of the number of times the chunk will be sent to node *j*, i.e., 1/(1 − *p_j_*), and the probability of node *j* having not received it. The value can, therefore, be calculated recursively with this formula:(14)$$E[n_{i,j}]\;=\;E[n_{i,j-1}]\;+\;\frac{1}{1-p_{j}}\cdot \;p_{j}^{E[n_{i,j-1}]}.$$

Using the same parameters employed in the previous subsection, the update time of this method can now be depicted: Figure 5a is obtained with *p_j_* = 1% for all the nodes, and Figure 5b with *p_j_* = 40%. The graphs show that the loss probability has a strong influence on the update time, as could be expected. In this case, the graphs keep on growing, even for *B* = 15: as the total time required by this method is lower, the time spent in the initial rounds is more significant than that of the *broadcast + unicast* method.

In this case, it is exactly the same to perform an initial broadcast round (*B* = 1) as just starting to sending the firmware chunks to the first node (*B* = 0), so the rest of the nodes will hear and store some of them. In addition, the optimal number of broadcast rounds is 0 or 1, i.e., making more than one round does not provide any benefit.

## 5. Simulation Tests and Results

The use of a simulation tool is seen as quite convenient to obtain results that can confirm the analytical ones. The advantage of simulations is that they are repeatable and can include a high number of nodes. This section describes the building of an ns-3 simulation environment, which is used to compare the different firmware update algorithms that have previously been proposed. The simulation only considers the firmware exchange phase (the *Sending of the checksum* and the *Checksum validation and firmware flashing* stages are not covered, since they are the same for every method). The tests and results with the real implementation will be reported in the next section.

### 5.1. Simulation Environment

The simulations include a single gateway, located in the center of the virtual scenario, and a number of nodes to be updated, distributed in a circular area around it (Figure 6). The distance to the gateway follows a uniform distribution between 0 and the maximum radius.

The setup uses the LoRa Phy layer developed in [28], with the Nagakami loss model [33], which matches the scenarios of interest. The Spreading Factor (SF) is fixed to 7, and the bandwidth of the LoRa channel is always 125 kHz. These two parameters are chosen because they are widely used in commercial deployments. A 1% duty cycle is used, as required by the LoRa Alliance in many cases.

The frames have a header of 23 bytes plus a payload of 192 bytes (by default). The size of the payload must be a multiple of 16 bytes, because this is the block size of the AES ciphering algorithm. In some tests, different frame sizes are employed.

As a summary, Table 1 includes all the parameters employed. Figure 7 shows the architecture of the simulation environment. On the one hand, it has the next input parameters: LoRa Spreading Factor, number of nodes, size of the firmware, number of iterations to be simulated, size of each firmware chunk, size of the ACKs, maximum radius, and number of broadcast rounds (only applicable in the broadcast methods). On the other hand, the simulation outputs are conformed by several csv files that include the peer coordinates on the scenario and the detailed results of the three FUOTA methods: the timestamp and RSSI (Received Signal Strength Indicator) of each frame exchange. The total time required for updating the whole project corresponds to the timestamp of the last frame.

The simulator uses a set of ns-3 libraries, including *LoRa Phy*, *LoRa End Device*, *LoRa Gateway*, and Delay/Loss Models. Furthermore, different C++ layers are included on top of ns-3: *LoRaSimGeo* (as scenario generator), *LoRaSimConst* (managing simulation constants), and *LoRaSimLogs* (for handling and saving logs and simulation results). *LoRaSim* is the main simulation layer, which handles the simulation parameters and launches each simulation iteration (for each FUOTA method), extracting results on the output files.

Finally, a set of Python scripts are used to manage the output files and build the graphs showing the results. The Pandas, Seaborn, and Matplotlib modules are used for that aim.

We first present a set of results for single runs of the OTA update of 10 nodes, with each of the three methods, using two values for the maximum radius: 400 m and 2 km. They are useful for obtaining insights about the behavior of each method. Then, the averaged results obtained with a battery of simulations will be presented in the following subsections, allowing us to devise more general conclusions.

### 5.2. Comparison between the Update Methods Using Single-Run Simulations

#### 5.2.1. Only Unicast Method

The results obtained in a scenario with a radius of 400 m are presented in Figure 8: the timestamp and the RSSI of the reception of each firmware chunk by the node are shown in (a), and the timestamp and the RSSI of the reception of each ACK by the gateway in (b). The red line represents the sensitivity of the receptor, i.e., the frames below that line are lost and retransmission is required.

As can be observed, in this case, almost all the frames are received correctly (the RSSI is above the sensitivity), so there are hardly any retransmissions. Consequently, the time for the update of each of the nodes is roughly the same. Another remark is that the difference between Figure 8a,b is minimal, which could be expected, considering that the channel is symmetrical and that a frame is always followed by an ACK.

The total update time of the project is 176,725 s, i.e., roughly 49 h, equivalent to less than 5 h per node. This time is significant, and is mainly imposed by the 1% duty cycle, which means that the air usage is slightly higher than 29 min.

Figure 9 reports the results of a scenario with a maximum radius of 2 km. As can be observed, some nodes (#4, #5, and #8, note that numbering starts with #0) are very close to the gateway, so they receive most of the frames upon the first attempt and their update times are shorter than those of other nodes such as, e.g., #2, #3, #6, and #9, which require many retransmissions. The time difference can be quite significant: from 35,664 s for node #6 to 21,258 s for node #8.

In this case, the total update time of the project is 294,613 s, i.e., 81.8 h (air usage of 49 min). The higher distance results in an increase of 66% in the update time with respect to the 400 m case.

#### 5.2.2. Broadcast + Unicast Method

The firmware update of 10 nodes, with a maximum radius of 400 m, is simulated using the *broadcast + unicast* method. The results presented in Figure 10a show the RSSI values seen by each node, i.e., during the broadcast phase (from 0 to 16,097 s), each frame appears 10 times, with the RSSI value corresponding to the receiving node. When this phase ends, a unicast one starts, in which the pending chunks of each node are completed (in this case, it takes 644 s to send 26 chunks to the nodes). As can be observed in Figure 10b, ACKs are sent by the nodes only in the unicast phase, in which the lost frames are sent individually.

The total time in this case is 16,775 s, i.e., 4.6 h. It represents 10.53% of the time required by the *only unicast* algorithm (see Figure 8): the update time is roughly divided by the number of nodes (*N* = 10 in this case). It should also be noted that the broadcast stage is faster than the unicast update of a single node, which required 17,672 s on average, because, in this case, no ACKs are required.

Next, another example illustrates what happens if the distance to the nodes is much higher (a maximum radius of 2 km). The results presented in Figure 11a show that, in this case, many frames arrive at the nodes below the sensitivity limit, so they are not received correctly. The broadcast stage lasts the same as that in the previous case: 16,097 s. But, in this example, the unicast stage is much longer: 120,316 s for the 10 nodes (see the ACKs Figure 11b).

The total time in this case is roughly 136,413 s, i.e., less than 38 h. It represents 46.3% of the time required by the *only unicast* method for the same distance (see Figure 9). The benefit is not so relevant in this case: a high number of chunks did not arrive correctly in the broadcast stage, so the unicast one is quite long. Nevertheless, it is shown that, in this case, a single initial broadcast round is able to halve the total update time.

#### 5.2.3. Only Broadcast Method

The third example presents the results of the *only broadcast* method in the small scenario (maximum radius of 400 m). The initial broadcast stage lasts 16,097 s (Figure 12a), as in the previous case. In the next stage, the gateway asks each node for the bitmap including the missing chunks. When it receives the response, the chunks are sent in broadcast frames. Figure 12b represents the RSSI of the received chunk if the node was missing that chunk. In this case, the first broadcast phase is very successful, so only 19 chunks are sent to the nodes (in broadcast frames) in the second stage.

The total time is 17,542 s, so it is slightly worse (4.5%) than that of the *broadcast + unicast* method. This difference is caused by the method followed in the final rounds: in the *broadcast + unicast* method, after a single request for the bitmap, all the pending chunks are sent and then the corresponding acknowledgement is received, which is relatively fast if the number of pending chunks is low. However, in the *only broadcast* method, a *SEND ME THE BITMAP* message has to be sent a number of times until an *ALL CHUNKS RECEIVED* message is sent by the node. Therefore, the number of messages is higher. However, if the number of pending chunks is high, the *only broadcast* method has the advantage of allowing any node to receive any pending chunk that is sent to the air in a broadcast frame, as we will see in the next example.

Another simulation is run using the big scenario (maximum radius 2 km). In this case, the broadcast phase lasts exactly the same time: one round which lasts 16,097 s (Figure 13a). The advantage with respect to the *broadcast + unicast* method can be clearly observed in the next stage: since the chunks are sent inside broadcast frames, other nodes different from the original destination may also receive and store them. In Figure 13b, we represent the RSSI of the frames received by the nodes that are missing the corresponding chunks. As can be observed in Figure 13b, at the beginning of this stage, a lot of nodes receive and store missing chunks which were requested by other nodes. However, when a node receives all the chunks, its color disappears from the graph.

This allows for a much faster completion of all the chunks for all the nodes. As a consequence of this advantage, the total update time is now 66,874 s. The second phase is reduced from 120,316 (*broadcast + unicast*) to 50,777 s (*only broadcast*). Overall, the *only broadcast* only takes 49% of the time required by the *broadcast + unicast* method, and it induces a reduction to 22.7% of the time required by the *only unicast* method.

### 5.3. Effect of the Size of the Scenario

To illustrate the effect of the size of the scenario, this subsection reports the results of a battery of simulations, with 10 nodes deployed in a circular area and a radius ranging from 100 m to 2 km. A fixed frame size of 215 bytes is used. The results are represented in terms of the total update time. Each of the presented values is the average of 40 simulations, and 95% confidence intervals are also included.

The *only unicast* method results are shown in Figure 14a: the increase in the radius results in a higher loss probability and an increase in the total time required for the update.

The benefit of the *broadcast + unicast* method is clear for short distances In Figure 14b, for example, for 100 m, the time is reduced to 9.2% of that required by the *only unicast* method. However, as the distance increases, it starts behaving similarly to the *only unicast* method: for 2 km, the time required by the *broadcast + unicast* is 35% of that required by the *only unicast* method. The reason is that the broadcast stage is not very successful, so most of the chunks are sent in unicast frames.

Finally, the *only broadcast* method (Figure 14c) is quite similar to the *broadcast + unicast* one for short distances (4.2% worse for 100 m), but it is able to maintain lower update times for long distances: for example, for 2 km, the time required by the *only broadcast* method is 65.7% of the time spent by the *broadcast + unicast* one.

### 5.4. Effect of Chunk Size

One question that can be considered is if there is an optimal value of the chunk size: on the one hand, a small size is expected to reduce the loss probability of a frame; on the other hand, a big size will reduce the overhead caused by LoRa headers. A battery of simulations is carried out in order to see whether this optimum exists. The simulations are run in a scenario with 10 nodes, using two values for the maximum radius: 400 m and 2 km. The header of the frames is 23 bytes long, and the payload ranges between 16 and 192 bytes, considering that it is ciphered with AES in blocks of 16 bytes. Each test is run 40 times. The 95% confidence intervals are also included.

In the small scenario (400 m, Figure 15), the three methods show similar behavior: the best option is always to use the longest payload (192 bytes plus the 23-byte header). Again, the *only unicast* method is the slowest one and, in this case, the two other methods present very similar results.

Things are different in the big scenario (2 km, Figure 16): as far as the *only unicast* method is concerned, the graph shows that bigger payloads (bigger chunks of firmware) are always beneficial for the update time. In this case, the best option is always to maximize the payload size.

However, in the case of the *broadcast + unicast* method, the tendency is just the opposite: if 39-byte frames are used, the update time is 80.4% of the time required if 215-byte frames are employed. This is caused by the high loss probability: if a frame is too long, the probability of not receiving it correctly is higher.

Something similar happens with the *only broadcast* method: in this case, it is also better to use small frames. In addition, there is a value of 135 bytes, which is the worst of all by a small difference (roughly 4.5%).

### 5.5. Effect of the Number of Nodes

In this subsection, we introduce a series of tests designed to assess the impact of varying the number of nodes updated. The frame size is fixed to 215 bytes, and two different values for the distance are employed: 400 m (Figure 17) and 2 km (Figure 18).

In the case of the *only unicast* method, it can be observed that the update time is (roughly) proportional to the number of nodes (this cannot be directly observed in the graphs, since they are in logarithmic scale).

The two methods that employ broadcast frames have a clear advantage, because a high number of nodes will potentially receive the chunks at the same time. In the *broadcast + unicast* method, it can be observed that the initial broadcast round saves a high percentage of the update time: if the scenario is small (400 m, Figure 17), it only requires 9.4% of the time required by the *only unicast* method for 10 nodes. When 150 nodes are used, the time required by the *broadcast + unicast* method is only 0.88% of the time used by the *only unicast* method (it is more than two orders of magnitude faster). If the scenario is higher (2 km, Figure 18), the saving is smaller: the first broadcast round is less effective, so the method behaves more closely to the *only unicast* approach. The time is 35% of the *only unicast* method for 10 nodes and 29.17% for 150 nodes.

As far as the *only broadcast* method is concerned, if the scenario is small (400 m), the results are slightly worse than those of the *broadcast + unicast* method: the initial broadcast round is quite effective, so many nodes receive the vast majority of the chunks in that moment. The *only broadcast* method is 4.2% worse for 10 nodes and 38.1% worse for 150 nodes.

However, the efficacy of the *only broadcast* method becomes distinctly evident in the big scenario (2 km): as the number of nodes increases, so does the benefit with respect to the *broadcast + unicast* method. With 10 nodes, the time is 65.7% of the one required by the *broadcast + unicast* method. For 150 nodes, the time is only 13.3%. The reason is that, with a higher number of nodes, the likelihood of a node receiving a chunk from any previous transmissions increases significantly.

This fits with the theoretical results (see Figure 4 and Figure 5): the *only broadcast* method is less sensitive to the number of nodes than the *broadcast + unicast* method: it is able to maintain low update times, even for high numbers of devices.

### 5.6. Effect of the Number of Broadcast Rounds

In Section 4, when analyzing the two broadcast methods, we discussed the possibility of having more than one initial broadcast round (*B* > 1) and concluded that, in some cases, it could save update time, in particular if the number of nodes is high, and also if the loss probability is significant. In this subsection, we present the results of a set of simulations run in a big scenario (2 km) with different numbers of initial broadcast rounds (*B* ranging from 1 to 15) and *N* = 10 or 20 nodes.

As far as the *broadcast + unicast* method is concerned (Figure 19a), the case *B* = 0 is equivalent to the *only unicast* method, as there is no initial broadcast round. The results in this case show that be best option is *B* = 1: making more rounds at the beginning does not improve the update time, which fits with the analytical results reported in Figure 4a.

In the case of the *only broadcast* method (Figure 19b), the behavior with *B* = 0 is exactly the same as that with *B* = 1, as explained in Section 4. It can be observed that this method is faster than the *broadcast + unicast* method. The optimal number of initial broadcast rounds is 1 in this case. When compared to Figure 5, it can be observed that the assumptions made in the analysis result in a slight underestimation of the update time: on the one hand, some simplifications were made; on the other hand, each simulated node has a different distance to the gateway, so the loss probability is not the same for every node.

### 5.7. Discussion about the Methods

As a first conclusion, the analysis and the simulations show that the update times can be significant, in some cases in the order of tens or hundreds of hours. This problem can be mitigated by the selection of the best-suited method.

There is a big difference between the *only unicast* method and the two ones in which broadcast transmission is employed. This fits with the results observed in the analytical section (see Figure 4 and Figure 5). The difference is usually significant: both broadcast methods can perform the firmware exchange using between 10 and 35% of the time required by the *only unicast* method. In small scenarios (up to 400 m), the broadcast methods are always one order of magnitude faster, and in some cases, this advantage can be of two orders of magnitude (150 nodes with a distance of 400 m). In big scenarios (2 km), the advantage of the broadcast methods is reduced: the *broadcast + unicast* method is able to complete the update in a time between 20 and 30% of the time, whereas the *only broadcast* method is usually faster. For example, with 150 nodes, it only requires 12.5% of the time required by the *broadcast + unicast* method. In conclusion, it seems clear that sending all the chunks in broadcast frames is the best option, also considering that there is an optimal value of the number of initial broadcast rounds. This also confirms the results obtained in the theoretical part (Figure 4 and Figure 5).

However, there are other considerations to be made, in order to establish the scenarios in which these conclusions are fully valid.

First, the two methods that send broadcast frames are valid if all the nodes run exactly the same firmware. Otherwise, the broadcast should be performed in groups, as this would reduce the benefit and complicate the implementation and control, as it requires a tight control of what has been received by each node.

The broadcast methods could also be used in LoRaWAN scenarios. For example, the firmware update presented in [8] showed an FUOTA protocol based on multicast frames. Its limitation, which would be overcome by the methods proposed in the present paper, is that it does not define a policy for retransmitting failed chunks. In contrast, our methods could enhance the whole update process in terms of speed and efficiency.

Finally, it should be considered that, although the conclusion that broadcast is the best option is valid in our case (LoRa with a constant spreading factor and channel bandwidth), it may not be applicable in other scenarios: if technology allowing an adaptive rate is employed, this can be different. For example, in 802.11, multicast frames, and by extension, broadcast ones, are typically transmitted at one of the *Basic Rates*, i.e., one of the rates that the Access Point (AP) has designated as mandatory for all its clients to support if they wish to connect [34]. This approach ensures that these frames are received by all clients, as these frames are not acknowledged at the 802.11 layer. Therefore, there would be cases in which the unicast option could be faster. As an example, if 10 nodes must receive new firmware, sending it individually in unicast frames at 54 Mbps will be faster than sending it once in broadcast frames at 1 Mbps.

## 6. Implementation and Tests with Real Hardware

In this section, we present some results of the OTA updates, measured in terms of total required time. A real setup is employed with low-end IoT devices based on the ARM-Cortex-M processor, a popular chip family which is optimized for low costs and energy efficiency. In this case, the *only unicast* version of the OTA procedure is implemented: in our work, there are nodes with different versions of the firmware because they have different sensors and implement various functions. The *only unicast* option is, therefore, implemented, as it simplifies the management of the firmware versions.

In addition to the architecture of the system, we will detail the update procedure as it happens in the node. In the last subsection, we will present some results of firmware updates performed with real hardware in our lab. In this case, the results will not be presented in terms of the total update time: a real setup has some limitations that do not appear in the simulation environment: number of devices, a high update time caused by duty cycle limitations, and interference, etc.

### 6.1. System Architecture and Hardware Implementation of the System Elements

The system architecture corresponds to a typical one employed in this kind of scenario. In contrast to LoRaWAN, in our case, the gateway implements some of the functions: for example, it can receive commands from the nodes and forward them to other ones, without requiring communication with the server. It has three main blocks (see Figure 20):

Node layer. The main element of the node is the Microchip WLR089U0 module, based on the ATSAMR34J18 LoRa integrated circuit. The PCB also includes a 32-bit ARM Cortex-M0+ processor with 256 KB of Flash and 40 KB of SRAM. It runs a firmware (roughly 150 KB) able to communicate with the gateway, and also with some sensors able to measure different magnitudes. An external flash non-volatile memory of 512 KB (SST25PF040C) is also added to the PCB via SPI protocol, which is the place where the new firmware is allocated. They can be considered as *Low-end Class* 2 devices [1]. No Operating System is used.Network layer. The gateway is an SOM (System On Module) based on Microchip ATSAMA5d27-wlsom1, which runs an embedded Linux as Operating System. It includes a WLR089U0 module connected via UART, working as the PAN Coordinator of the LoRa network. In addition, the SOM platform has an Ethernet and a Wi-Fi card, which provide connectivity to the server. A Mosquitto MQTT broker provides connectivity between the gateway and the server.Server layer. It integrates different elements: a back-end server which interacts with a database and a web application where the user can control the gateway and the nodes to launch the OTA updates.

The nodes are associated to the gateway following a star topology. In normal conditions, the gateway periodically requests new samples from them. These samples may include the data measured by the DALI driver (temperature and power consumption, etc.) or by other sensors attached to the ATEX luminaire. The format of the LoRa frames is detailed in Appendix A. When an OTA update is to be performed, this process can be interrupted to leave all the available airtime for the update.

Two photographs of the node, inside an ATEX enclosure, are presented in Figure 21: the PCB, the DALI driver, the emergency module, and the backup batteries can be seen in Figure 21a. A detail of the other side is presented in Figure 21b, where the LED stripe can be observed, in addition to the enclosure of a presence node and the LoRa antenna.

Two photographs of the gateway are presented in Figure 22. As can be observed, it includes the SOM at the bottom and two PCBs (for simultaneously operating in two channels).

### 6.2. Detailed Update Procedure

The memory map of the internal flash memory is shown in Figure 23. The bootloader, stored at the beginning, handles the boot of the device. It is also responsible for receiving, checking, and writing the updated firmware into the flash memory. Its size is 10 KB approximately.

There is a 4 KB section of shared space between the bootloader and the application’s firmware, allowing for communication between them. It is used for booting purposes, with three modes: (a) *application booting*, the normal one; (b) *FUOTA booting*, after an FUOTA process; and (c) *serial-update booting*, after an update via serial port, which is only used for updating the firmware of the LoRa PCB of the gateway.

During the FUOTA process, the application firmware handles the saving of the new firmware into the external memory; later, it checks the correctness of the SHA256 hash. If correct, it communicates to the bootloader that a new firmware has been stored by using the shared section; finally, it resets the device. After the reset, the bootloader, running in the *FUOTA booting* mode, checks the firmware stored on the external memory, and, if it is correct, it overwrites the internal flash memory (also with SHA256 verification); finally, the bootloader is launched again, this time in the *application booting* mode.

### 6.3. Tests of the Unicast Method for the OTA Update with Real Hardware

A laboratory setup us used to carry out several tests of the OTA update. As a first example, Figure 24 shows the RSSI of each frame during a complete OTA procedure, with 812 chunks of 197 bytes each. The distance between the gateway and the node is 2 m, without any obstacle between them. Both the gateway and the node are out from their ATEX enclosures to reduce the signal attenuation.

In this case, considering that the tests are run in an isolated place, the frames are sent in a continuous mode, not considering the 1% duty cycle. The total required time is 807 s (roughly 13 min and a half).

As can be observed, the vast majority of the frames arrive on the first attempt (yellow dots), although some arrive on the second one (green dots). The RSSI is always around −49 dBm, with reasonably stable behavior.

One fact should be noted: in the simulation tests, frame loss occurred solely due to the RSSI level falling beneath the sensitivity threshold of the receiver. In the tests with real hardware, there may be other causes: interference with equipment in the same frequency bands and limitations of the hardware, etc. This is the reason why some frames are lost in spite of the big margin between the RSSI level (−49 dBm) and the sensitivity of the receptor (−125 dBm).

Another test is carried out between a gateway in the lab and a node on the upper floor. In this case, there is a significant attenuation level, mainly caused by the slab between the floors. As can be observed in Figure 25, the RSSI level ranges between −85 and −110. In addition, it can be observed that many frames require two attempts, and some even require three. However, the algorithm is solid and can complete the firmware transfer in 2450 s (more than 40 min).

### 6.4. Discussion about the Implementation

Different lessons have been learned with the implementation, which will be discussed in this subsection. Regarding the steps that are common to the three methods, it is important to remark that the size of the firmware must be kept low, for two reasons: first, considering the tight constraints imposed by airtime limitations, the time required for updating a project grows with the number of chunks; and second, the hardware of the IoT device also poses some space constraints: the new firmware must fit in the internal flash of the chip and also in the external flash non-volatile memory. These elements must be well-dimensioned from the beginning, considering that a hardware modification is not an option in many of these projects.

There is another related consideration: if the nodes have different connected elements (e.g., different sensors), several versions of the firmware will be required. In these cases, two options can be considered: first, compiling all the functionalities and using them selectively; and second, using preprocessor directives and performing conditional compilation. In principle, the best option for systems with resource constraints is the second one, as the resulting binary is smaller. This is clearly the best option when the firmware is updated using unicast frames. However, if a broadcast method is used, it could make sense to compile all the functionalities and generate a single version of the firmware, as this would increase the scalability.

The most critical moment of the firmware update is the flashing of the new version, the copy of the binary between the two flash memories, and the reboot with the new code. This must be performed very carefully with several mechanisms to ensure that no problem arises, as this would render the node inoperative.

Finally, one comment regarding the *only unicast* method: it has been shown that although it is not as scalable as the other two, it performs well with a quite simple control mechanism (“stop and wait”). This can be an advantage in noisy environments or if the number of firmware versions is significant.

## 7. Conclusions and Future Research Directions

This paper presented, analyzed, and compared three methods for performing the firmware exchange required by an Over The Air update of a project including a number of IoT nodes connected via LoRa. After presenting the three methods, an analytical expression of the required update time was obtained for each of them.

The *only unicast* method is the one that privileges control versus the update time: it allows an easier implementation and a simplified update, using a “stop and wait” policy. If a time reduction is desired, it was shown that an initial broadcast round (or more than one) and then a unicast completion of the missing chunks can significantly reduce the required time: this is the *broadcast + unicast* method. Furthermore, an additional improvement can be included, sending the missing chunks in broadcast frames as well (the *only broadcast* method). This can reduce the required time even more, especially if the distance and the number of nodes are high. Finally, it was shown that there is an optimal number of initial broadcast rounds.

The methods were simulated in ns-3 varying different parameters, namely, the distance between the gateway and the nodes, the number of nodes, and the size of the payload. The advantages of each method were highlighted. The results showed that significant time reductions can be obtained by using the proper method: in some cases, if the number of nodes to be updated is high, the update time can be reduced by two orders of magnitude. Finally, one of the methods was implemented in real hardware, and some tests were run with it. The lessons learned from the implementation were explained in detail.

It appears evident that broadcasting as many frames as possible is the optimal choice in our current scenario. This approach is particularly effective in the case of LoRa, where a fixed rate is employed. However, it is important to note that this strategy may not be suitable for all situations. For instance, if a project includes nodes with different firmware versions, the benefits would be mitigated. Furthermore, when using technology that supports adaptive rates, the dynamics could change: if broadcast frames are transmitted at lower rates, the unicast option can become potentially faster. Therefore, the choice of broadcast or unicast should be performed carefully, based on the specific technology and scenario at hand.

As far as future research directions are concerned, more scenarios can be simulated and other methods can be implemented. For example, alternatives to the “stop and wait” approach could be explored, i.e., “go back N” or even “selective ACK.” This would be valid for the *only unicast* method and for the second stage of the *broadcast + unicast* method. Furthermore, different combinations of the three methods could be explored.

More simulations could be performed, including interfering nodes. Other effects could then be observed. For example, it is possible that the use of smaller chunks provides a better performance in these cases.

Considering the increasing interest in the use of machine learning for IoT environments [35], it would also be interesting to have a tool that takes, as an input, the characteristics of the scenario (the number of nodes, the groups of nodes that share the same firmware version, the distances, the size of the firmware, the energy consumption, and the background traffic, etc.) and generates an output specifying the best-suited method and best parameters for performing a firmware update on it. This tool could be trained using simulations or real results and improve its output automatically.

Although the present paper is focused on the firmware exchange stage, the security of the previous stages could be improved, adding new layers based on blockchain [36].

In addition, the update methods could be tested in the context of IoT projects based on other communication protocols such as NBIoT [37] and Zigbee [38], or even proprietary ones such as Enhanced ShockBurst (ESB) running in nRF24L Series chips [39].

Finally, other KPIs apart from the update time could be estimated and compared, such as the energy cost incurred by each method. Although energy is not a problem in our specific use case, it can be relevant in other situations.

## Figures and Tables

**Figure 1 sensors-24-02104-f001:**
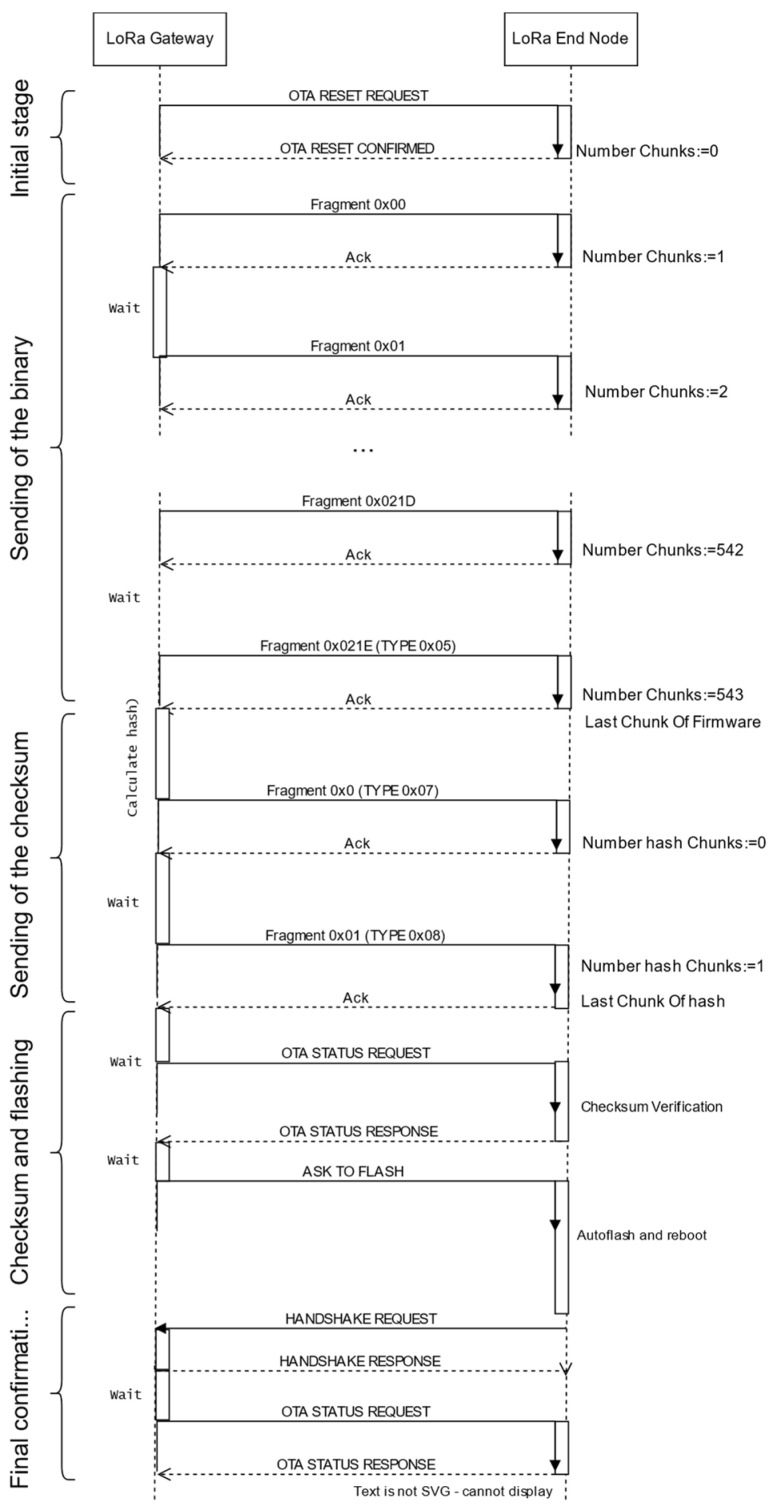
Stages of the *only unicast* method.

**Figure 2 sensors-24-02104-f002:**
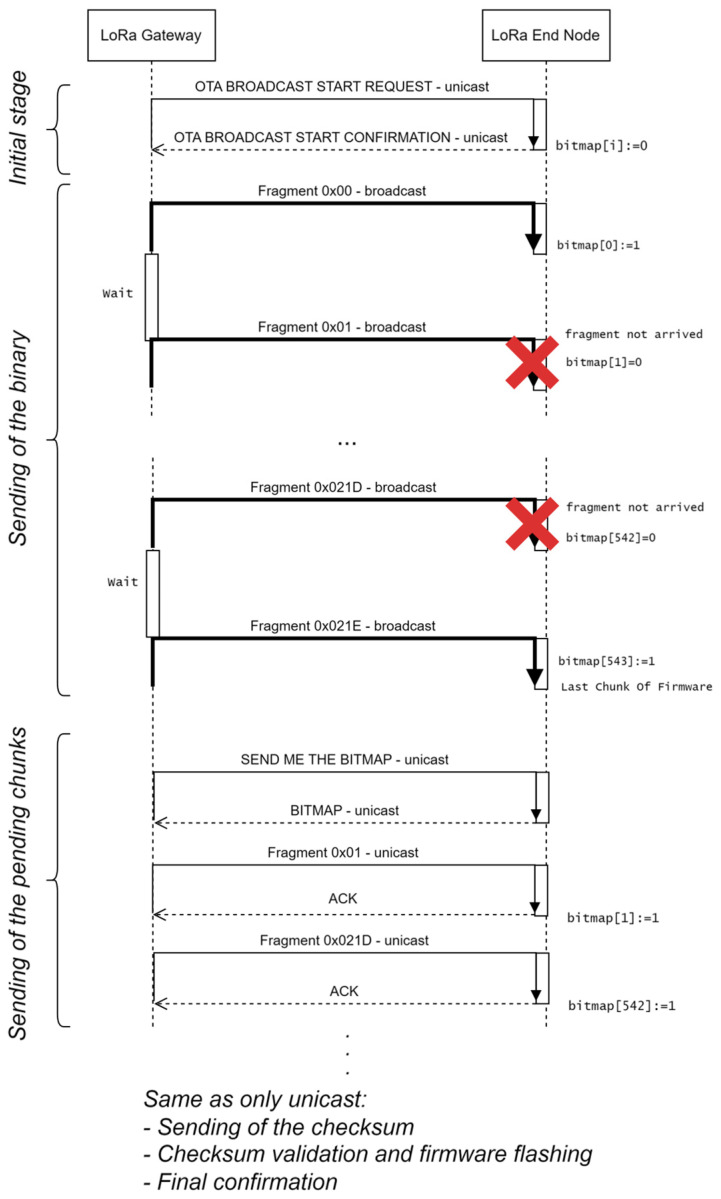
Detail of stages (a), (b), and (c) of the *broadcast + unicast* method. The red crosses correspond to lost frames.

**Figure 3 sensors-24-02104-f003:**
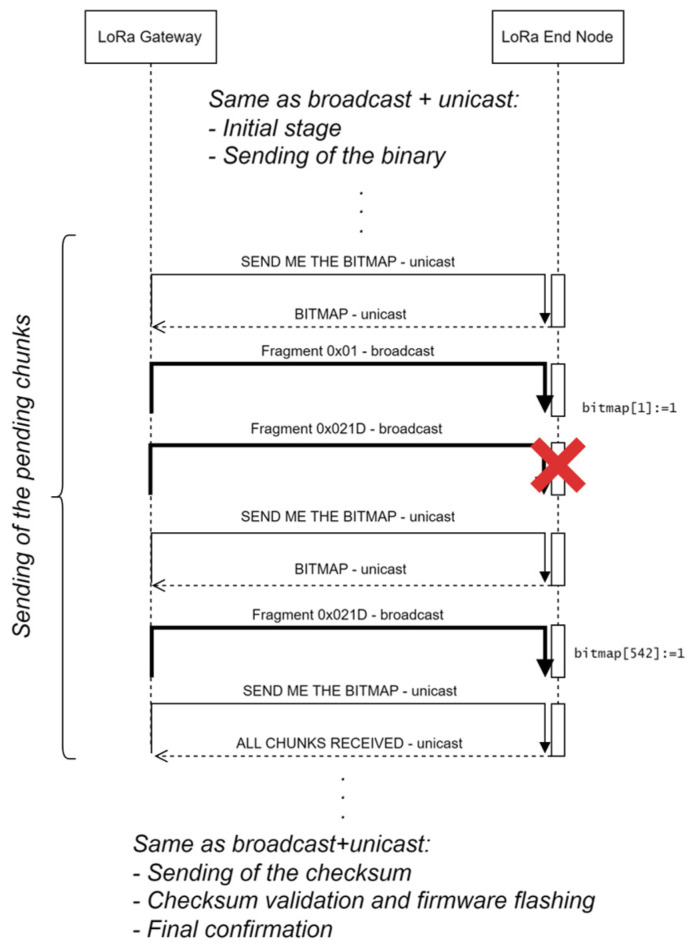
Detail of stage (c) of the *only broadcast* method.

**Figure 4 sensors-24-02104-f004:**
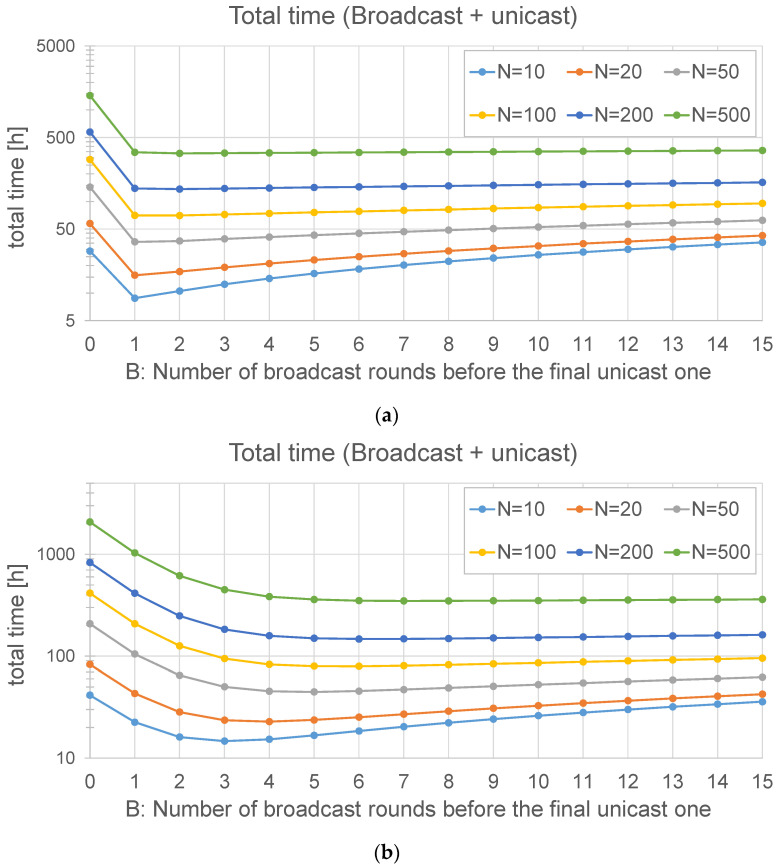
*Broadcast + unicast* method: total update time for different values of the number of nodes and the number of broadcast rounds: (**a**) loss probability = 1% and (**b**) loss probability = 40%. Note that the *Y* axis is in logarithmic scale, and that the ranges of (**a**,**b**) are different.

**Figure 5 sensors-24-02104-f005:**
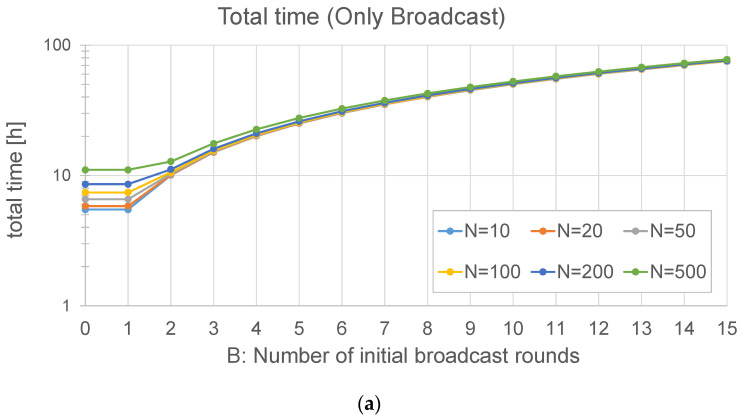
*Only broadcast* method: total update time for different values of the number of nodes and the number of initial broadcast rounds: (**a**) loss probability = 1% and (**b**) loss probability = 40%. Note that the *Y* axis is in logarithmic scale, and that the ranges of (**a**,**b**) are different.

**Figure 6 sensors-24-02104-f006:**
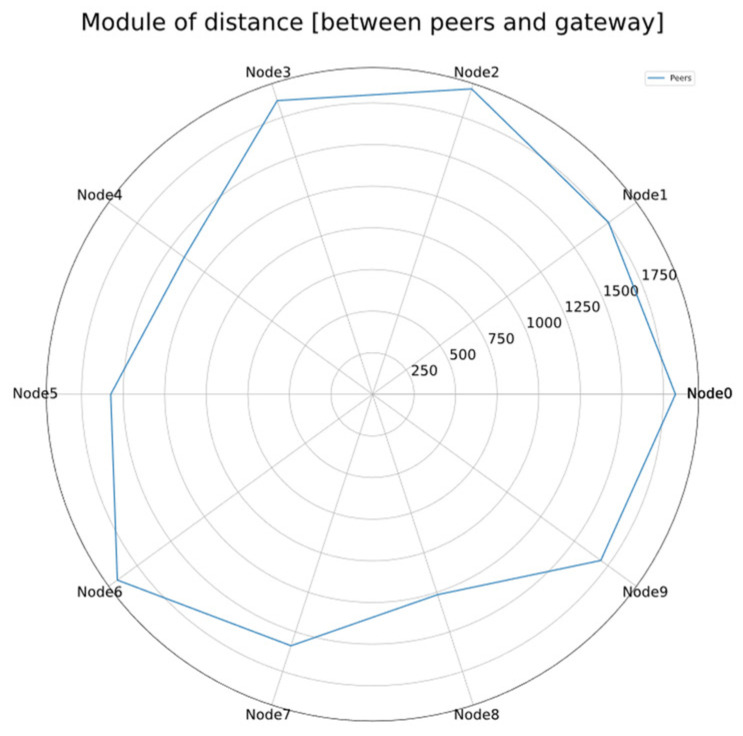
Circular scenario with 10 peers and a maximum radius of 2000 m.

**Figure 7 sensors-24-02104-f007:**
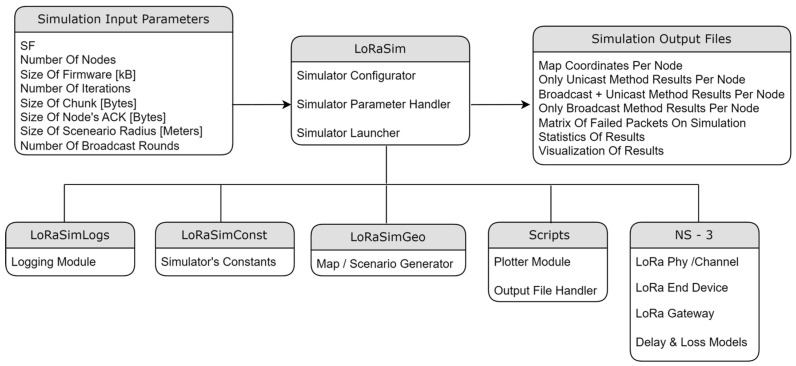
LoRa simulation environment (on top of ns-3).

**Figure 8 sensors-24-02104-f008:**
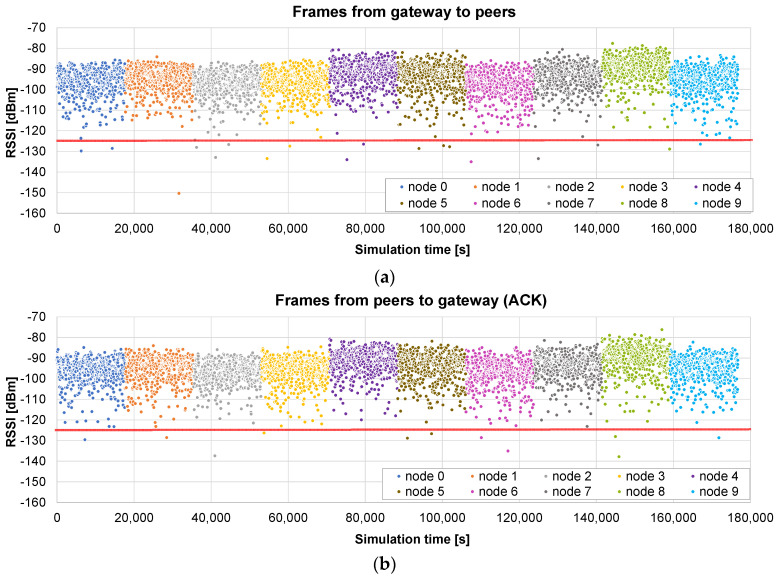
*Only unicast* firmware update of 10 peers with a maximum radius of 400 m: (**a**) RSSI of the frames received by each node, containing the firmware chunks; and (**b**) RSSI of the ACK frames received by the gateway. The red line corresponds to the sensitivity of the receiver.

**Figure 9 sensors-24-02104-f009:**
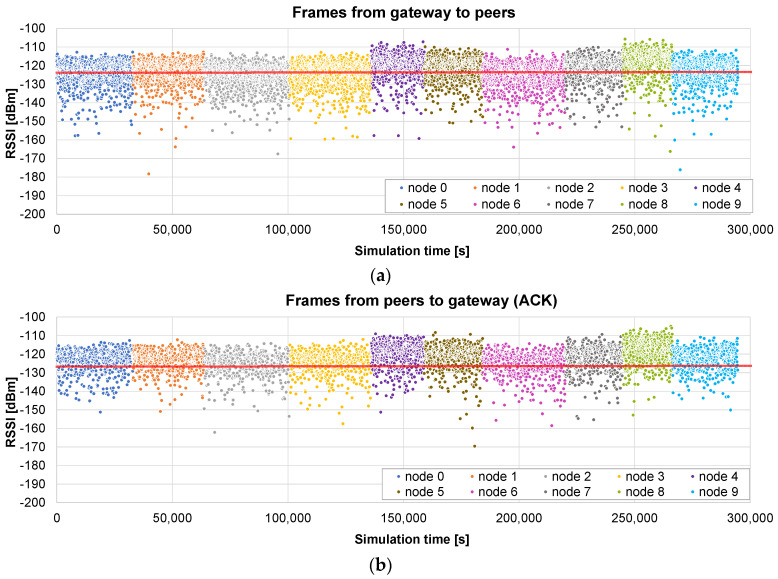
*Only unicast* firmware update of 10 peers with a maximum radius of 2 km: (**a**) RSSI of the frames received by each node, containing the firmware chunks; and (**b**) RSSI of the ACK frames received by the gateway. The red line corresponds to the sensitivity of the receiver.

**Figure 10 sensors-24-02104-f010:**
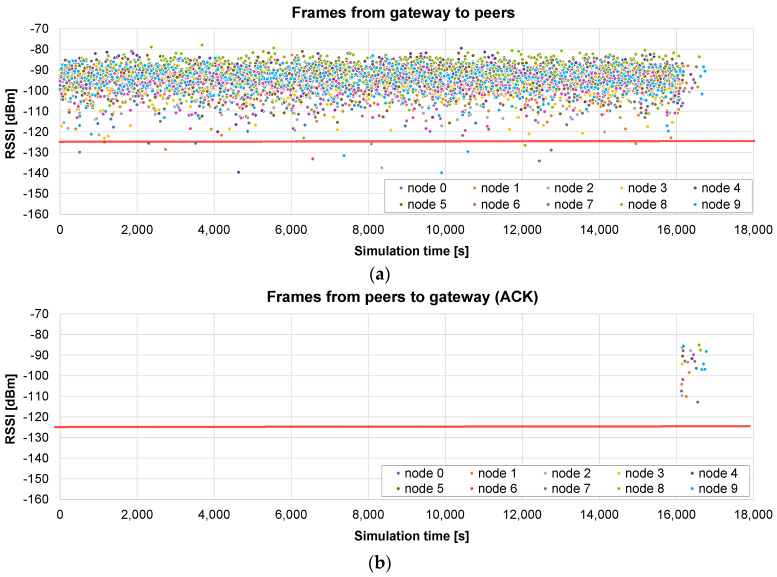
*Broadcast + unicast* firmware update of 10 peers with a maximum radius of 400 m: (**a**) RSSI of the frames received by each node, containing the firmware chunks; and (**b**) RSSI of the ACK frames received by the gateway. The red line corresponds to the sensitivity of the receiver.

**Figure 11 sensors-24-02104-f011:**
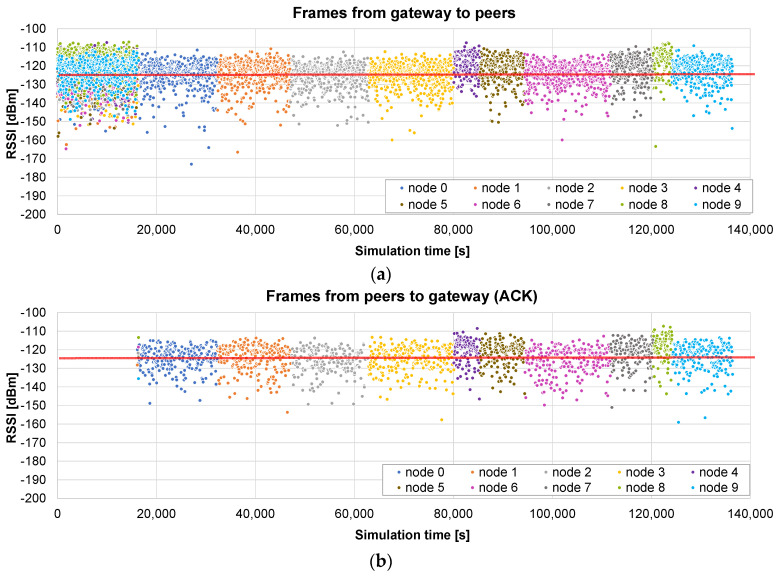
*Broadcast* + *unicast* firmware update of 10 peers with a maximum radius of 2 km: (**a**) RSSI of the frames received by each node, containing the firmware chunks; and (**b**) RSSI of the ACK frames received by the gateway. The red line corresponds to the sensitivity of the receiver.

**Figure 12 sensors-24-02104-f012:**
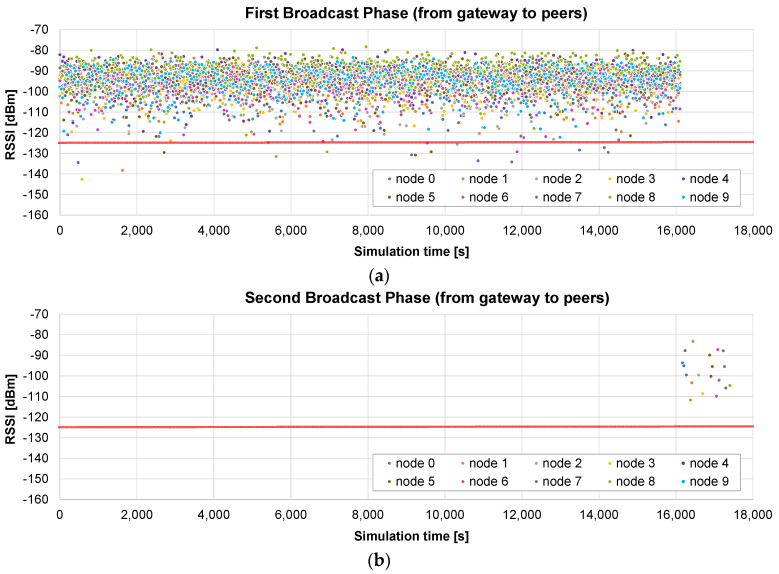
*Only broadcast* firmware update of 10 peers with a maximum radius of 400 m: (**a**) RSSI of the frames sent by the gateway, containing the firmware chunks; and (**b**) RSSI of the frames received by the nodes which are missing them. The red line corresponds to the sensitivity of the receiver.

**Figure 13 sensors-24-02104-f013:**
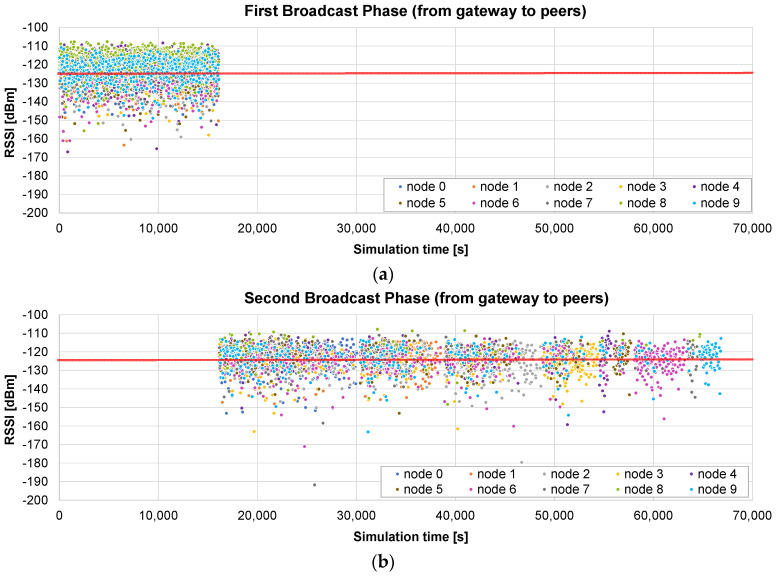
*Only broadcast* firmware update of 10 peers with a maximum radius of 2 km: (**a**) RSSI of the frames sent by the gateway, containing the firmware chunks; and (**b**) RSSI of the frames received by the nodes which are missing them. The red line corresponds to the sensitivity of the receiver.

**Figure 14 sensors-24-02104-f014:**
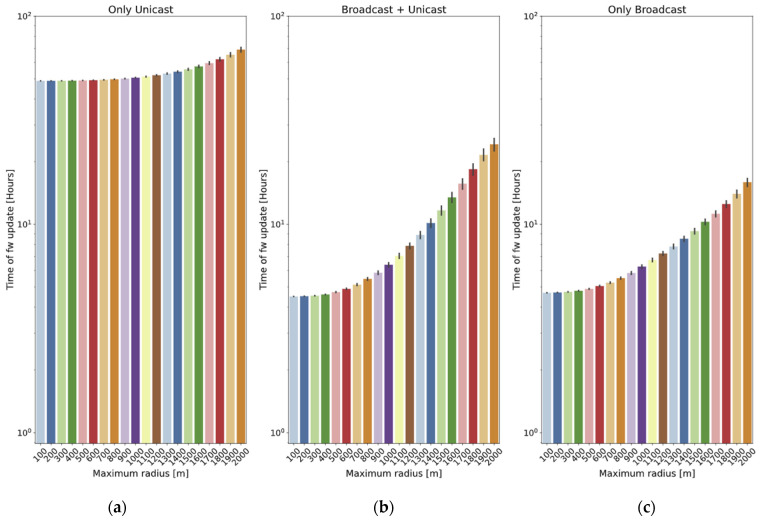
Effect of the radius of the scenario on the total update time: (**a**) *Only unicast* method; (**b**) *Broadcast + unicast* method; and (**c**) *Only broadcast* method. Note that the *Y* axis is in logarithmic scale.

**Figure 15 sensors-24-02104-f015:**
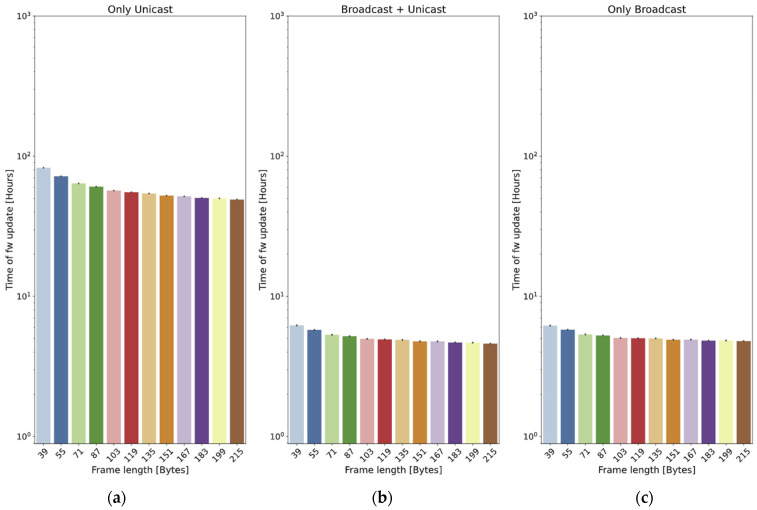
Effect of payload size on update time. Maximum radius 400 m: (**a**) *Only unicast* method; (**b**) *Broadcast + unicast* method; and (**c**) *Only broadcast* method. Note that the *Y* axis is in logarithmic scale.

**Figure 16 sensors-24-02104-f016:**
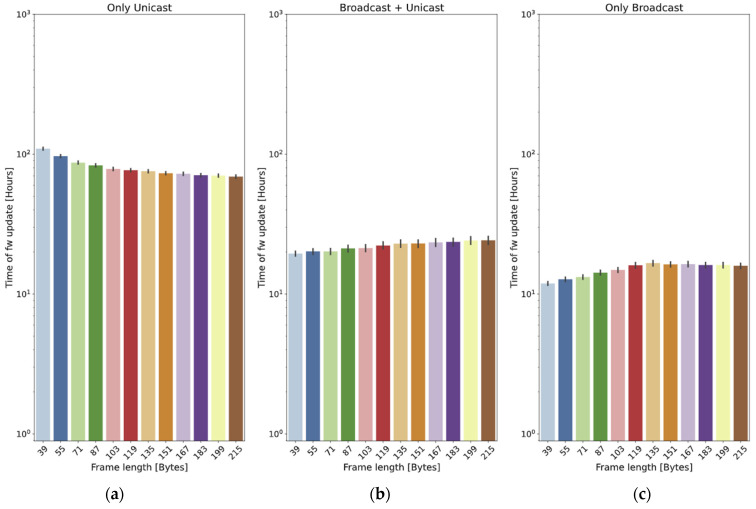
Effect of payload size on update time. Maximum radius 2 km: (**a**) *Only unicast* method; (**b**) *Broadcast + unicast* method; and (**c**) *Only broadcast* method. Note that the *Y* axis is in logarithmic scale.

**Figure 17 sensors-24-02104-f017:**
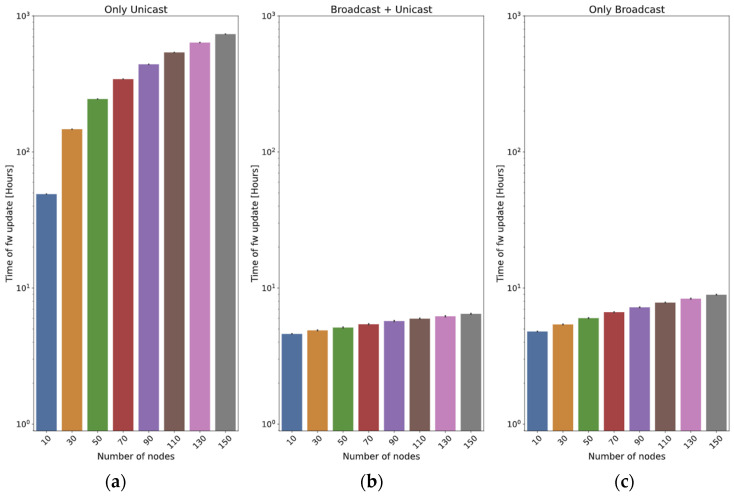
Effect of the number of nodes on update time. Maximum radius 400 m: (**a**) *Only unicast* method; (**b**) *Broadcast + unicast* method; and (**c**) *Only broadcast* method. Note that the *Y* axis is in logarithmic scale.

**Figure 18 sensors-24-02104-f018:**
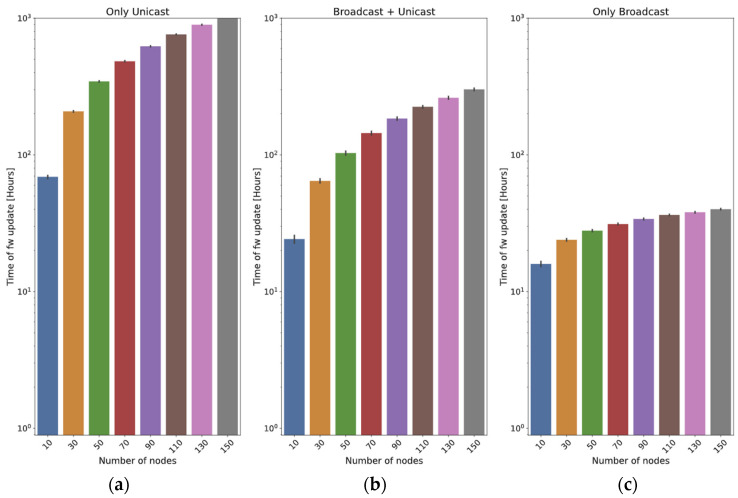
Effect of the number of nodes on update time. Maximum radius 2 km: (**a**) *Only unicast* method; (**b**) *Broadcast + unicast* method; and (**c**) *Only broadcast* method. Note that the *Y* axis is in logarithmic scale.

**Figure 19 sensors-24-02104-f019:**
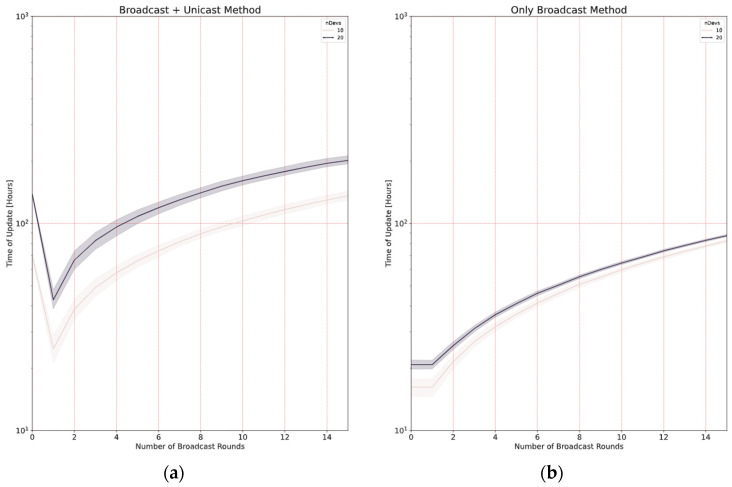
Effect of the number of initial broadcast rounds (*B*) on the update time, with 10 and 20 nodes, and a maximum radius of 2 km: (**a**) *Broadcast + unicast* method and (**b**) *Only broadcast* method. Note that the *Y* axis is in logarithmic scale.

**Figure 20 sensors-24-02104-f020:**
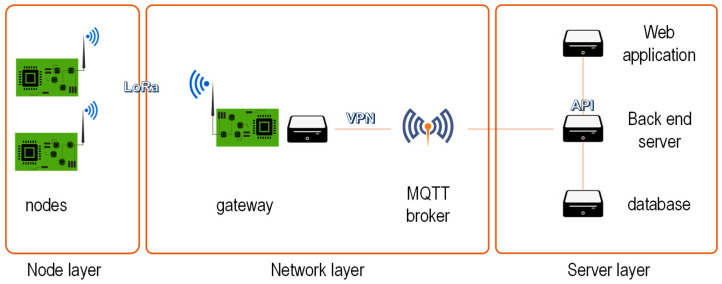
Architecture of the system.

**Figure 21 sensors-24-02104-f021:**
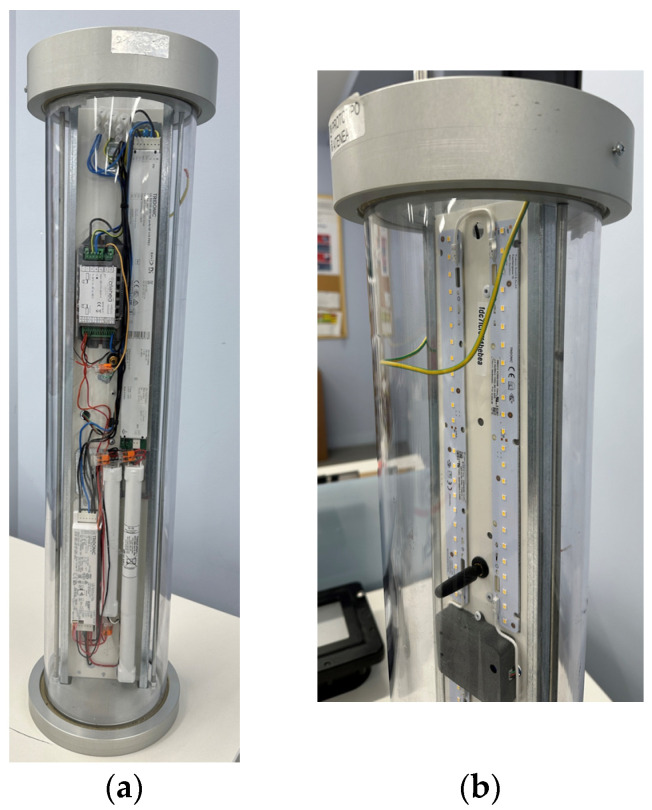
Images of the node: (**a**) the side where the PCB, the DALI driver, and other elements are placed; and (**b**) detail of the opposite side: LED stripe, LoRa antenna, and presence sensor enclosure.

**Figure 22 sensors-24-02104-f022:**
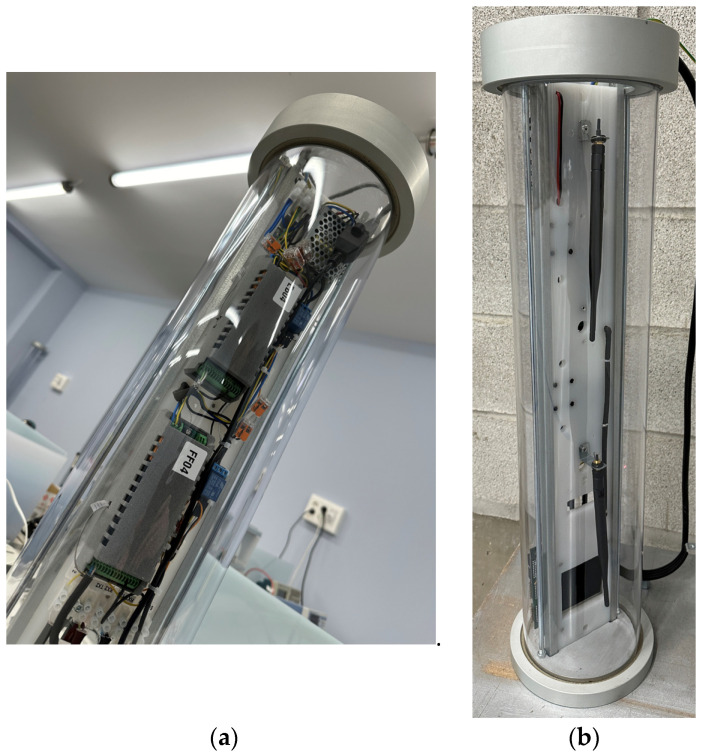
Images of the gateway: (**a**) the side where two PCBs are placed; and (**b**) opposite side: two LoRa antennas.

**Figure 23 sensors-24-02104-f023:**
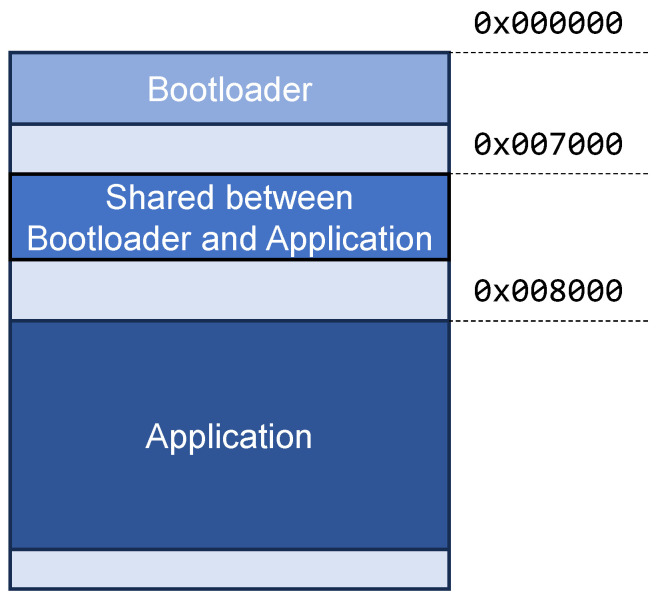
Memory map of the flash memory.

**Figure 24 sensors-24-02104-f024:**
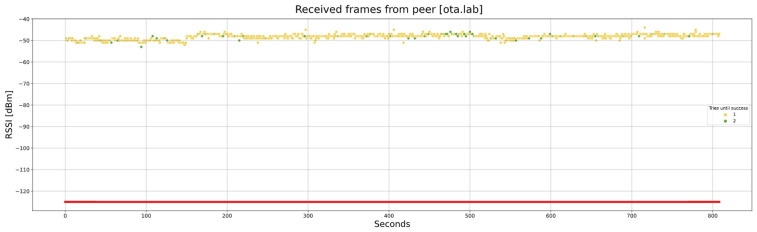
Unicast OTA procedure, with a distance of 2 m and no obstacles: RSSI of each frame. The red line corresponds to the sensitivity of the receiver.

**Figure 25 sensors-24-02104-f025:**
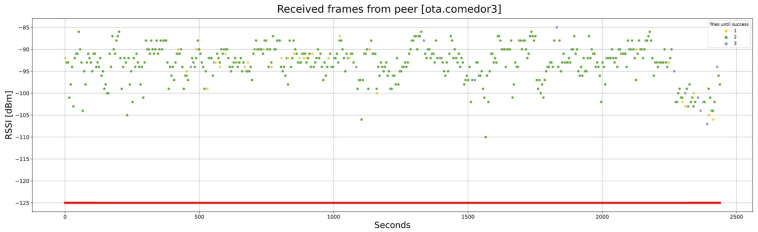
Unicast OTA procedure with a significant distance and obstacles: RSSI of each frame. The red line corresponds to the sensitivity of the receiver.

**Table 1 sensors-24-02104-t001:** Simulation parameters.

Parameter	Range/Values
Circle scenario radius	From 100 to 2000 m
Number of nodes to update (*N*)	10 to 150 nodes
Number of initial broadcast rounds (*B*) (only for the broadcast methods)	1 by default
Default size of a frame containing a firmware chunk	215 bytes (23 header + 192 payload)
Firmware size	Fixed to 100 kB
Spreading factor	Fixed to SF7
Bandwidth of the LoRa channel	Fixed to 125 kHz
Channel	868 MHz
Sensitivity of the LoRa receptors	Fixed to −125 dBm
Duty cycle	1%
Propagation model	Log Distance (ns3::LogDistancePropagationLossModel)
Loss model	Nagakami (ns3::NakagamiPropagationLossModel)

## Data Availability

Data are contained within the article.

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
