# Peer review of "Firmware Updates over the Air via LoRa: Unicast and Broadcast Combination for Boosting Update Speed"

_sensors, 2024, doi:10.3390/s24072104_

Round 1

Reviewer 1 Report

Comments and Suggestions for Authors

This paper investigates three methods for performing firmware exchanges required by Over The Air updates in a project with numerous nodes. The methods include unicast, broadcast + unicast, and broadcast-only approaches. Analytical expressions for update times are derived for each method, with the unicast method prioritizing control over update time. The paper demonstrates that incorporating initial broadcast rounds followed by unicast completion can significantly reduce update times. Additionally, sending missing chunks in broadcast frames further reduces update times, particularly for distant nodes. Simulation in ns-3 with varied parameters confirms the advantages of each method, with some achieving substantial time reductions, up to two orders of magnitude in certain scenarios. Furthermore, one method is implemented in real hardware, yielding promising results. The paper highlights that maximizing broadcasting is optimal in their scenario, especially in LoRa technology with fixed rates, but notes potential limitations, such as mixed firmware versions or adaptive rate support, which may warrant careful consideration between broadcast and unicast options based on specific technology and scenario requirements. Future directions are also provided and are actually quite interesting for other researches.

As evident from the summary above, the work is quite extensive and complex. The paper is well written, clear, and technically sound.

The topic perfectly fits the journal's scope.

The experiments are well explained to allow their replicability. NS-3 parameters seems reasable and well chosen (including the wireless model which is a realistic one).

The authors did both simulations and real experiments.

All in all, it seems that the authors did a meticulous work.

I can find only a couple of very simple suggestions to improve the paper; otherwise, the paper is already ready as it is.

1) Sometimes the authors write meters and sometimes m; please be uniform. Also, I prefer 10% rather than 10 % with the blank between 10 and %. (but I might be wrong, so these are just suggestions but not mandatory at all)

2) Related work section and papers in bibliography could be extended with some more recent papers.

For instance (but the authors can find more):

https://www.mdpi.com/1424-8220/21/19/6488

https://www.mdpi.com/2079-9292/12/13/2952

https://ieeexplore.ieee.org/abstract/document/9217759?casa_token=X-58-7dCCG0AAAAA:QD0YO0fUgCOIjHfNHMcVOTIFNPedlatyn5xYfAhNXdW59bgCYoZ2ylDaT3z8hhDoAHYhuiqS

Reviewer 2 Report

Comments and Suggestions for Authors

IMPROVEMENTS/GRAMMAR ERRORS/TYPOS:

  • In the Abstract, “To complete the study, the unicast method has been implemented and some results are also presented” this sentence seems to break the flow. Please make sure the flow reads well. 
  • Line 13 (Abstract) "employing" the least amout of time ==> it may be better say in the least amount of time or spanning the least amount of time.
  • Can you elaborate on duty cycle limitations? Why can only 1% of airtime be used? Can you eloborate on the calculation on line 403?
  • In line 709, did you mean 88% rather than 0.88% ? 
  • Could you please double check Eqn. 1 ? First, could you elaborate how the deduction become possible? Second, on line 405 the formula is indicated differently, how did that happen?
  • On line 430, "approximation of considering" --> please revise the sentence to improve clarity
  • Can you please explain Figure 4? Why does the time cost stabilize after some broadcast rounds? Why is the behavior different from Figure 5 which keeps slightly increasing with the number of broadcast rounds?
  • Line 536, etc. please use present tense in the Simulations section, not future tense.
  • I did not notice, until Section 6, that there was a real-world implementation in this paper. You have only mentioned simulations at the beginning of Section 5. Please rephrase the first paragraph of Section 5 to indicate that you first performed simulations which are explained in this section, and the implementation results will be reported in the next section.
  • In Section 6 it would be nice to include plots or tables having the time cost on the y axis, as this is the figure of merit in your study.
  • Some of the short paragraphs in the Introduction can be merged together, to improve readability with fewer interrupts.
  • Below each equation in the Analytical Evaluation (Section 4), could you please elaborate on (explain in words) the terms and how do you derive one from the other (for example how do you derive Eqn. 7 from Eqn. 6) ? 

Reviewer 3 Report

Comments and Suggestions for Authors

1. Authors have used “in our project” in the manuscript. Authors should make it clear that it should be “experiment or work” instead of project.

2. There is a mismatch between the abstract and the content of the manuscript.

3. The contributions should be reframed in a brief manner.

4. “broadcast” implementation is missing.

5. In Fig. 2 and Fig. 3, the stages representation is missing.

6. English language should be improved extensively in terms of grammar and sentence formation.

7. There is lots of redundant content and the organization of the manuscript should be improved.

8. Figure 20 is a general architecture, what is the contribution of authors?

9. The results shown in Figures 4 and 5 should be compared with the results of the existing methods.

10. The manuscript is too lengthy.

11. Overall, the work presented in the manuscript seems to be a undergraduate level student project. It needs major improvements in terms of quality and novelty.

Comments on the Quality of English Language

English language should be improved extensively in terms of grammar and sentence formation.

Reviewer 4 Report

Comments and Suggestions for Authors

Clarify the specific criteria used to determine the "best way" to deliver firmware fragments to IoT nodes. Providing a detailed evaluation framework or set of metrics will enhance the transparency and reproducibility of the study's findings.

Specify the exact characteristics of the low-end IoT nodes and their hardware resources. Detailed information about the hardware constraints will provide valuable context for readers to understand the challenges and limitations addressed by the proposed OTA update methods.

Provide a clear rationale for the selection of the three different methods for OTA updates. Justifying why these specific methods were chosen and how they address different aspects of the update process will enhance the paper's credibility and relevance.

Include more detailed descriptions of the analytical and simulation methodologies used to explore the different OTA update methods. Providing insights into the simulation setup, parameters, and assumptions will allow readers to better understand the validity and applicability of the results.

Present a comprehensive analysis of the results obtained from both analytical exploration and extensive simulations. Comparing and contrasting the performance of each OTA update method under various scenarios will provide valuable insights into their effectiveness and applicability in different contexts.

Discuss potential limitations or constraints associated with the implementation of the unicast method and its impact on the overall findings. Addressing any challenges or shortcomings encountered during the implementation process will enhance the paper's credibility and help readers better interpret the results.

Offer recommendations or insights for future research directions based on the findings of the study. Identifying areas for further investigation or potential improvements to the OTA update methods will contribute to the ongoing advancement of IoT device management techniques.

Please avoid citing sources that were published before to 2019. Cite current research that are really pertinent to your topic. The study also lacks sufficient citations. Another critical step is to compare the topic of the article to other relevant recent publications or works in order to widen the research's repercussions beyond the issue. Authors can use and depend on these essential works while addressing the topic of their paper and current issues.

Bakhshi, T., Ghita, B., & Kuzminykh, I. (2024). A Review of IoT Firmware Vulnerabilities and Auditing Techniques. Sensors24(2), 708.

Heidari, A., Navimipour, N. J., & Unal, M. (2022). Applications of ML/DL in the management of smart cities and societies based on new trends in information technologies: A systematic literature review. Sustainable Cities and Society, 104089.

Yan, Shu-Rong, et al. "Implementation of a Product-Recommender System in an IoT-Based Smart Shopping Using Fuzzy Logic and Apriori Algorithm." IEEE Transactions on Engineering Management (2022).

Comments on the Quality of English Language

 Minor editing of English language required.

Round 2

Reviewer 4 Report

Comments and Suggestions for Authors

Well revised.